# ImageNet3D: Towards General-Purpose Object-Level 3D Understanding

**Wufei Ma[1], Guofeng Zhang[1], Qihao Liu[1], Guanning Zeng[2],**
**Adam Kortylewski[4,5], Yaoyao Liu[6], Alan Yuille[1]**
[1]Johns Hopkins University    [2]Tsinghua University    [4]University of Freiburg
[5]Max Planck Institute for Informatics    [6]University of Illinois Urbana-Champaign

## Abstract

A vision model with general-purpose object-level 3D understanding should be capable of inferring both 2D (*e.g.*, class name and bounding box) and 3D information (*e.g.*, 3D location and 3D viewpoint) for arbitrary rigid objects in natural images. This is a challenging task, as it involves inferring 3D information from 2D signals and most importantly, generalizing to rigid objects from unseen categories. However, existing datasets with object-level 3D annotations are often limited by the number of categories or the quality of annotations. Models developed on these datasets become specialists for certain categories or domains, and fail to generalize. In this work, we present ImageNet3D, a large dataset for general-purpose object-level 3D understanding. ImageNet3D augments 200 categories from the ImageNet dataset with 2D bounding box, 3D pose, 3D location annotations, and image captions interleaved with 3D information. With the new annotations available in ImageNet3D, we could (i) analyze the object-level 3D awareness of visual foundation models, and (ii) study and develop general-purpose models that infer both 2D and 3D information for arbitrary rigid objects in natural images, and (iii) integrate unified 3D models with large language models for 3D-related reasoning. We consider two new tasks, probing of object-level 3D awareness and open vocabulary pose estimation, besides standard classification and pose estimation. Experimental results on ImageNet3D demonstrate the potential of our dataset in building vision models with stronger general-purpose object-level 3D understanding. Our dataset and project page are available here: `https://imagenet3d.github.io`.

## 1    Introduction

General-purpose object-level 3D understanding requires models to infer both 2D (*e.g.*, class name and bounding box) and 3D information (*e.g.*, 3D location and 3D viewpoint) for arbitrary rigid objects in natural images. Correctly predicting these 2D and 3D information is crucial to a wide range of applications in robotics [1, 2] and general-purpose artificial intelligence [3, 4, 5]. Despite the success of previous learning-based approaches [6, 7], embodied or multi-modal LLM agents with stronger 3D awareness will not only reason and interact better with the 3D world [8, 9], but also alleviate certain key limitations, such as shortcut learning [10] or hallucination [11, 12].

Despite the importance of object-level 3D understanding, previous datasets in this area were limited to a very small number of categories [13, 14, 15] or specific domains, such as autonomous driving [16, 17] or indoor furniture [18]. Subsequent works then focused on developing specialized models that excel at 3D tasks for the categories and domains considered in these datasets. While these specialized models are found useful for certain downstream applications, they fail easily when generalizing to novel categories. It is largely understudied of how to develop unified 3D models that are capable of inferring 2D and 3D information for all common rigid objects in natural images.

38th Conference on Neural Information Processing Systems (NeurIPS 2024) Track on Datasets and Benchmarks.

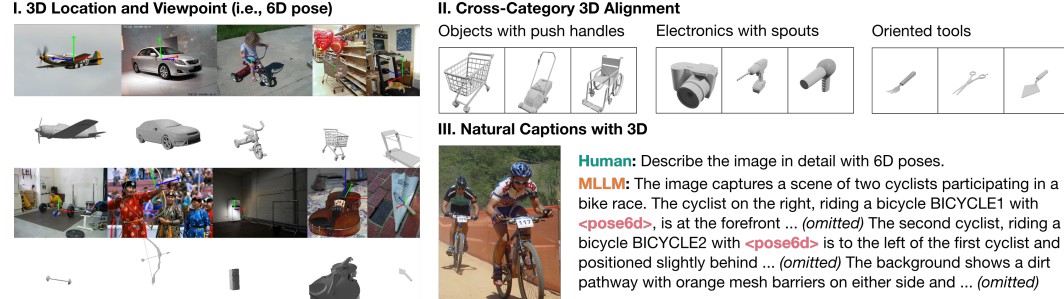

I. 3D Location and Viewpoint (i.e., 6D pose)

II. Cross-Category 3D Alignment

Objects with push handles    Electronics with spouts    Oriented tools

III. Natural Captions with 3D

**Human:** Describe the image in detail with 6D poses.
**MLLM:** The image captures a scene of two cyclists participating in a bike race. The cyclist on the right, riding a bicycle BICYCLE1 with <pose6d>, is at the forefront ... *(omitted)* The second cyclist, riding a bicycle BICYCLE2 with <pose6d> is to the left of the first cyclist and positioned slightly behind ... *(omitted)* The background shows a dirt pathway with orange mesh barriers on either side and ... *(omitted)*

Figure 1: **Overview of ImageNet3D data and annotations.** ImageNet3D provides **3D location and viewpoint (*i.e.*, 6D pose)** for more than 86,000 objects. We also annotate **cross-category 3D alignment** for the 200 rigid categories in ImageNet3D. Lastly we generate **image captions interleaved with 3D information** to integrate unified 3D models with large language models.

In the following, we consider two types of unified 3D models. **(i) Pretrained vision encoders with object-level 3D awareness.** Vision encoders from DINO [19], CLIP [20], Stable Diffusion [21], etc. are pretrained with self-supervised or weakly-supervised objectives. By learning a 3D discriminative representation, these vision encoders can be integrated into vision systems and benefit downstream recognition and reasoning. *While these encoders are found useful for 3D-related dense prediction tasks [22], their object-level 3D awareness remains unclear.* **(ii) Supervised 3D models.** By training on a large number of diverse data with 3D annotations, these models may achieve a stronger robustness and generalization ability. **However, there has been a lack of large-scale 3D datasets with a wide range of rigid categories, which constrains us from developing large unified 3D models for rigid objects or study the generalization and emerging properties of these models.**

In this work, we present ImageNet3D, a large dataset for general-purpose object-level 3D understanding. We extend 200 categories from ImageNet21k [23] with 2D bounding box and 6D pose annotations for more than 86,000 objects. To facilitate research on the two problems introduced above, ImageNet3D incorporates three key designs (see Figure 1). **(i) A large number of categories and instances.** ImageNet3D presents 2D and 3D annotations for a large number of object instances from a wide range of common rigid object categories found in natural images, as opposed to previous datasets focusing on specific categories and domains (see Table 1). This allows us to train and evaluate large unified 3D models capable of inferring both 2D and 3D information for arbitrary rigid objects. **(ii) Cross-category 3D alignment.** We align the canonical poses of all 200 categories based on semantic parts, shapes, and common knowledge, as shown in Figure 3. This is crucial for models to benefit from joint learning from multiple categories and to generalize to unseen categories, while omitted in previous datasets [24]. **(iii) Natural captions with 3D information.** We adopt a GPT-assisted approach [6] and produce image captions interleaved with 3D information. These captions will be valuable assets to integrate unified 3D models with large language models [25, 26] and perform 3D-related reasoning from natural images and language.

With the three key designs and new 3D annotations collected, ImageNet3D distinguishes itself from all previous 3D datasets and facilitates the evaluation and research of general-purpose object-level 3D understanding. Besides standard classification and pose estimation as studied in previous works [13, 24], we further consider two new tasks, probing of object-level 3D awareness and open-vocabulary pose estimation. Experimental results show that with ImageNet3D, we can develop general-purpose models capable of inferring 3D information for a wide range of rigid categories. Moreover, baseline results on ImageNet3D reveal the limitations of current 3D approaches and present new problems and challenges for future studies.

## 2 Related Works

**Datasets with 3D annotations.** Previous datasets with 3D annotations have led to significant advancements of 3D models for 3D object detection [29, 30] and pose estimation [31, 32, 14]. However, most existing datasets are limited to a small number of categories [13, 14, 15] or specific domains, such as autonomous driving [16, 17] or indoor furniture [18]. ObjectNet3D [24] extends the

| Dataset | Images | # categories | # instances | Annotations |
|---|---|---|---|---|
| PASCAL3D+ (2014) [13] | Real | 12 | 12,000 | 6D pose |
| ObjectNet3D (2016) [24] | Real | 100 | 57,000 | 6D pose |
| CAMERA25 (2019) [27] | Synthetic | 6 | 1,000 | 3D bbox |
| REAL275 (2019) [27] | Real | 6 | 24 | 3D bbox |
| Objectron (2021) [28] | Real | 9 | 18,000 | 3D bbox |
| Wild6D (2022) [14] | Real | 5 | 2,000 | 3D bbox |
| ImageNet3D (ours) | Real | 200 | 86,000 | 6D pose, captions, object visual quality, cross-category 3D alignment |

Table 1: **Comparison between ImageNet3D and previous datasets with 3D annotations.** Previous datasets are limited by the number of rigid categories [13, 28, 14] or the quality of the annotations [24], constraining the development of large unified 3D models for general-purpose 3D understanding.

number of categories but the quality of the annotations constrains us from developing large unified 3D models. Our ImageNet3D largely extends the number of categories and instances, improves the annotation quality, and presents other crucial annotations such as cross-category 3D alignment and natural captions interleaved with 3D information. ImageNet3D allows us to develop unified 3D models for general-purpose 3D understanding and facilitates studies on new research problems, such as probing of object-level 3D awareness and open-vocabulary pose estimation.

**Category-level pose estimation.** Our work is closely related to category-level pose estimation, where a model predicts 3D or 6D poses for arbitrary instances from certain rigid categories. Previous works have explored keypoint-based methods [31] and 3D compositional models [32, 33]. However, these approaches limited their scopes to a small number of categories, and as far as we know, there were no attempts to develop large unified models for all common rigid categories. We further consider open-vocabulary pose estimation where models generalize to similar but novel categories. This topic has also been discussed in recent parallel works [34, 35] but [35] was limited to synthetic data rendered with photorealistic CAD models.

**3D awareness of visual foundation Models.** Recent work demonstrates the significant capabilities of large-scale pretrained vision models in 2D tasks [20, 36, 37, 38, 21], suggesting robust 2D representations. Beyond benchmarking the semantic and localization capabilities of visual backbones [39, 40, 41, 42, 43, 44], Banani et al. [22] studied the 3D awareness of these 2D models using trainable probes and zero-shot inference methods. However, their exploration was limited to only two basic aspects of 3D understanding – single-view surface reconstruction and multi-view consistency – due to absence of large datasets with 3D annotations. We further analyze the 3D awareness of visual models and provide a more comprehensive understanding of their progress in learning about the 3D structure of the world, demonstrating the significance of our proposed ImageNet3D.

## 3 ImageNet3D Dataset

ImageNet3D dataset aims to facilitate the evaluation and research of general-purpose object-level 3D understanding models. Besides 6D pose annotations for more than 86,000 objects from 200 categories, we annotate meta-classes, cross-category 3D alignment, and natural captions interleaved with 3D information as demonstrated in Figure 1. We start by presenting our dataset construction in Section 3.1. Then in Section 3.2 we introduce the necessity of cross-category 3D alignment and how it is achieved in our dataset. Lastly, we provide details on our image caption generation in Section 3.3.

### 3.1 Dataset Construction

**Overview.** We choose the ImageNet21k dataset [23] as our data source, as it provides a large and diverse set of images with class labels. We start by annotating 2D bounding boxes for the object instances in the images. We adopt a machine-assisted approach for 2D bounding box annotations, where a Grounding DINO model [45] is used to produce 2D bounding boxes prompted with the category label. The bounding box annotations are then filtered and improved by human evaluators. Next, we collect 3D CAD models as representative shapes for each object category from Objaverse [46].

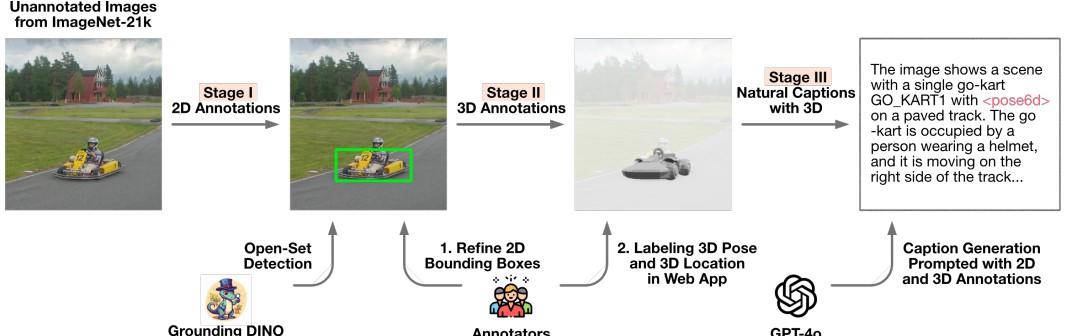

Figure 2: An overview of our ImageNet3D dataset creation pipeline.

The CAD models are carefully aligned based on their semantic parts and provide canonical poses for 6D pose annotations. For 3D annotations, we recruit a total of 30 annotators to annotate 6D poses for the objects, as well as the scene density and object visual quality. Lastly we generate natural captions interleaved with 6D poses with GPT-4o. An overview of our data generation pipeline is visualized in Figure 2.

**Object categories.** Our goal is to provide 3D annotations for all common rigid categories in real world. To achieve this, we carefully examine previous 2D and 3D datasets for image classification [23], object detection [47, 29, 28], and pose estimation [24, 27]. We choose the categories that are rigid, have well-defined shapes with certain variance, and have enough number of images available, which leads to the 200 categories in ImageNet3D. For detailed discussions on the choice of categories, please refer to Section A.1. Moreover, to leverage existing research in the field, we adopt the 100 categories and raw images from ObjectNet3D [24] and largely extend the number of categories and instances. As one of our goals is to improve the quality of 3D annotations, we only take unannotated images from ObjectNet3D, and all 3D annotations on these images are our original work. In Section D, we perform human evaluation on the annotation qualities in ObjectNet3D [24] and our ImageNet3D.

**Annotator recruitment.** We recruit 30 annotators for data annotation. To improve the quality of the collected data, each annotator must complete an onboarding stage before starting. The onboarding stage includes training sessions where we present detailed instructions of various annotations and proper ways to handle boundary cases. Additionally, each annotator must annotate sample questions and meet the accuracy threshold to qualify for subsequent work. Please refer to Section A and Section C regarding our annotator guidelines, training sessions, and ethics statement.

**Data collection.** We develop a web-based tool for data annotation so annotators can easily access the platform without local installation. A screenshot of our annotation tool is shown in Figure 5. For each object in ImageNet3D, the annotator needs to annotate the following. **(i) 3D location and 3D viewpoint (i.e., 6D pose):** For more intuitive annotating, the 3D location is parameterized as a combination of 2D location and distance of the object. The 3D viewpoint is defined as the rotation of the object with respect to the canonical pose of the category, and represented by three rotation parameters, azimuth, elevation, and in-plane rotation (see Figure 6). **(ii) Density of the scene:** A binary label indicating if the scene is dense with many objects from the same category close to each other. **(iii) Visual quality of the object:** A categorical label with one of the four options: good, partially visible, barely visible, not visible. We refer the readers to Section C where we provide links to our annotation guidelines and instructions.

## 3.2 Cross-Category 3D Alignment

As explained in Section 3.1, the 3D viewpoint of an object is defined as the rotation of the object with respect to the canonical pose of this category. However, in previous datasets such as ObjectNet3D [24], canonical poses from different categories are not necessarily "aligned". From the canonical poses depicted in Figure 4, the parts where the pencils "write" or the paintbrushes "paint" are pointing to different directions, and the spouts of faucets and kettles are also mis-aligned.

**As we scale up the number of categories in 3D-annotated datasets, having cross-category 3D alignment is a crucial design for the study of general-purpose object-level 3D understanding.**

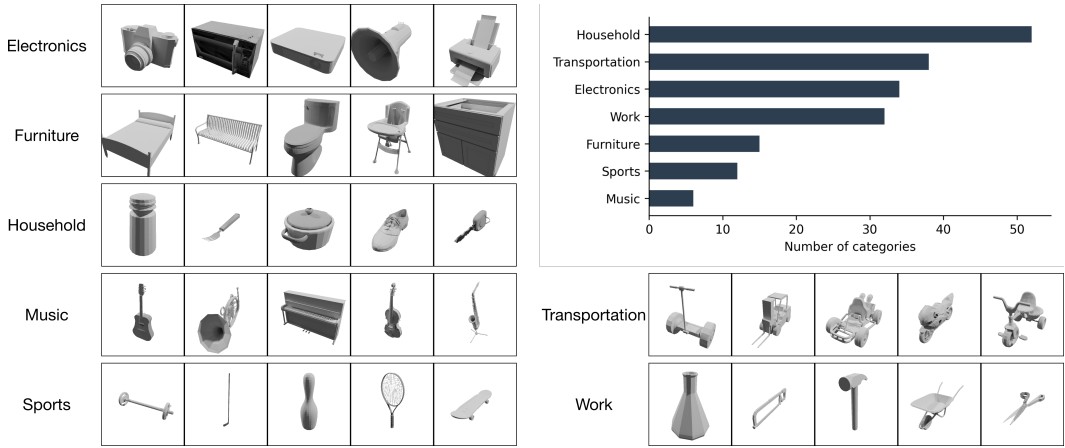

Figure 3: **Meta classes and cross-category 3D alignment.** We align the canonical poses of all 200 categories based on semantic parts, shapes, and common knowledge. This is crucial for models to benefit from joint learning from multiple categories and to generalize to novel categories.

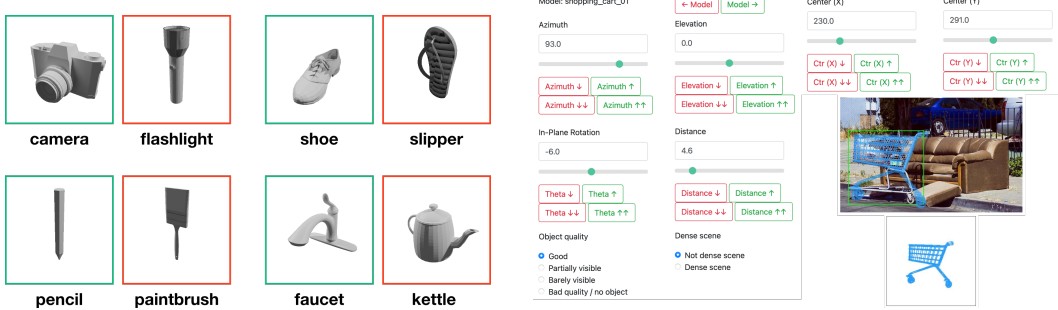

Figure 4: **Mis-aligned canonical poses in Ob-jectNet3D [24].**

Figure 5: **Screenshot of our web app for data annotation.**

While objects from different categories have their unique characteristics, certain semantic parts are often shared between multiple categories, such as the wheels of "ambulances" and "forklifts" or push handles of "shopping carts" and "hand mowers". Correctly aligning the canonical poses will (i) allow models to utilize the semantic similarities between parts of different categories and exploit the benefits of joint learning from multiple categories, and (ii) generalize to novel categories by inferring 3D viewpoints from semantic parts that the model has seen from other categories during training.

Therefore, we manually align the canonical poses of all 200 categories in ImageNet3D. Specifically, we consider the following three rules. **(i) Semantic parts:** categories sharing similar semantic parts, such as wheels, push handles, or spouts, should be aligned. **(ii) Similar shapes:** categories sharing similar shapes, such as fans, Ferris wheels, and life buoys, should be aligned. **(iii) Common knowledge:** certain categories are pre-defined with a "front" direction from common knowledge, such as "refrigerator", "treadmill", or "violin".

## 3.3 Natural Captions with 3D Information

An important application of general-purpose object-level 3D understanding models is to integrate them with large language models (LLMs) and benefit downstream multi-modal reasoning. This would largely improve the 3D-awareness of multi-modal large language models (MLLMs) and improve 3D-related reasoning capabilities, such as poses [9] and distances [48]. Previous approaches integrated segmentation or human pose modules with MLLMs [25, 26] and demonstrate strong multi-modal reasoning abilities.

To integrate general-purpose 3D understanding with existing MLLMs, we present image captions interleaved with 3D information. As shown in Figure 1, our captions provide a detailed description

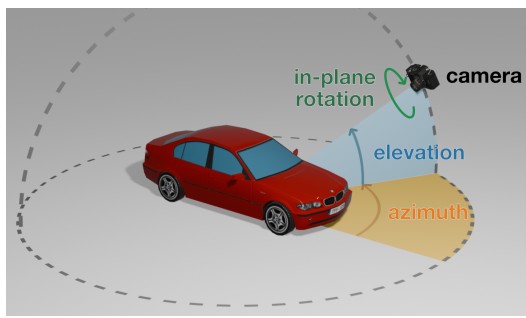

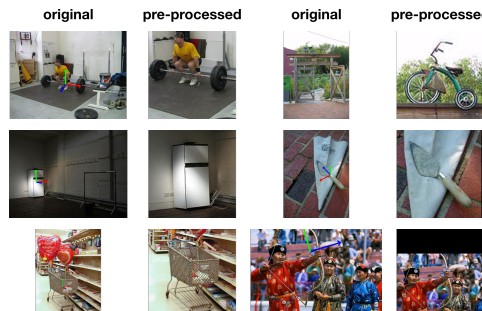

Figure 6: Illustration of the 3D viewpoint parameterization.

Figure 7: Qualitative examples of our data pre-processing.

of the image, object appearances and locations, as well as mutual relations. Moreover, for objects with 3D annotations, we add a special <pose6d> token as a reference to our 2D and 3D annotations for this object. To generate these image captions with 3D information, we adopt a GPT-assisted approach [6] and feed 2D and 3D annotations to the model via the textual prompts. Then GPT-4v is used to integrate these information and produce a coherent image caption interleaved with 3D information. Please refer to Section A.3 for details on caption generation as well as our GPT-4v prompts.

## 4 Tasks

With the new data and annotations available in ImageNet3D, we hope to push forward the evaluation and research of general-purpose object-level 3D understanding. We consider two new tasks, probing of 3D object-level awareness 4.1 and open-vocabulary pose estimation 4.2, besides joint image classification and category-level pose estimation 4.3. We further other standard computer vision tasks, such as image classification and object detection, and report the full performance in our dataset page.

### 4.1 Linear Probing of Object-Level 3D Awareness

Recent developments of large-scale pretraining have yielded visual foundation models with strong capabilities. Self-supervised approaches such as MAE [49] and DINO [19] provide strong and generalizable feature representations that benefit downstream recognition and localization. When jointly trained with language supervision, CLIP features [20] demonstrate transferability to a wide range of multi-modal tasks. Moreover, foundation models for specific tasks, *e.g.*, MiDaS [50] for depth estimation, also show impressive capabilities when applied to arbitrary images.

Are these visual foundation models object-level 3D aware? Can these feature representations distinguish objects from different 3D viewpoints or retrieve objects from similar 3D viewpoints? A parallel work [22] found that certain foundation models have better 3D awareness despite trained without 3D supervision. However, they focused on low-level tasks such as depth estimation and part correspondence. It remains unclear if these visual foundation models are object-level 3D aware and produce 3D discriminative object representations.

In this task, we aim to evaluate the object-level 3D awareness of visual foundation models by linear probing the frozen feature representations on 3D viewpoint classification task. This is because models with superior object-level 3D awareness would produce 3D discriminative features that help to classify the viewpoints correctly. Compared to low-level tasks such as depth estimation and part correspondence, object-level 3D awareness is directly associated with high-level scene understanding that is crucial to downstream recognition and reasoning in robotics and visual question answering.

**Task formulation.** We evaluate object-level 3D awareness by linear probing the frozen feature representations on 3D viewpoint classification task. Specifically, three linear classifiers are trained with respect to each of the three parameters encoding 3D viewpoint. To ensure that the neural features encode rich information about the target object with 3D annotations, we adopt a data pre-processing step where we crop and resize the image based on the 2D and 3D annotations (see Figure 7). Following the linear probing setting on ImageNet1k [51], we apply grid search to a range

of hyperparameters, such as learning rate, pooling strategy, and training epochs, and select the best performance achievable with the frozen backbone features.

**Evaluation.** To jointly evaluate the classification results on three viewpoint parameters, we adopt the **pose error** given by the angle between the predicted rotation matrix and the groundtruth rotation matrix [31]

$$\Delta(R_{\text{pred}}, R_{\text{gt}}) = \frac{\|\text{logm}(R_{\text{pred}}^\top R_{\text{gt}})\|_{\mathcal{F}}}{\sqrt{2}} \tag{1}$$

Based on the pose errors, we compute **pose estimation accuracy**, which is the percentage of samples with pose errors smaller than a pre-defined threshold.

## 4.2 Open-Vocabulary Pose Estimation

Existing 3D models for pose estimation [31, 32, 14] or object detection [30, 29] focused on scenarios where object images and 3D annotations from the target categories are available at training time. These models fail easily when generalizing to novel categories that posses similar semantic parts with categories that the models are trained on. A recent study [34] investigated the open-vocabulary pose estimation problem from synthetic data rendered with photorealistic CAD models. However, the synthetic dataset demonstrates limited variations in both object appearances and image backgrounds, while our ImageNet3D provide 3D annotations on real images from a wide range of rigid categories to study this problem.

How can 3D models generalize to novel categories? Intuitively models may utilize semantic parts that are shared between novel categories and categories that are seen during training. As demonstrated in Figure 8, models may generalize 3D knowledge learned from cars (*i.e.*, sedans and SUVs) to fire trucks based on the wheels and body of vehicles, or from hand barrows to shopping cars based on the push handles. Additionally, open-vocabulary pose estimation models may utilize large-scale 2D pre-training data or vision-language supervision and learn useful semantic information. For instance, after seeing 2D images of people riding a bicycle and a tricycle, models would learn to align the semantic parts and generalize from bicycles to tricycles. Lastly we provide detailed descriptions of object shape, part structure, and how humans interact with these objects for all categories in ImageNet3D (see Section A.1). Models may utilize such information and learn transferable features that generalize to novel rigid categories.

**Task formulation.** We split the 200 categories in ImageNet to 63 common categories for training and 137 categories for open-vocabulary pose estimation. Models may utilize additional 2D data for pretraining but may be only trained on 3D annotations from the 63 common categories. During testing time, models have access to our annotated category-level captions besides the testing images. For complete lists of categories used for training and open-vocabulary pose estimation, please refer to Section C.

**Evaluation.** Following standard pose estimation tasks [13, 24], we report *pose estimation accuracy* and median *pose error* (Eq. 1) on testing data from novel categories that are unseen during training.

## 4.3 Joint Image Classification and Category-Level Pose Estimation

For joint image classification and category-level pose estimation, a model first classifies the object and then predicts the 3D viewpoint of the object. A prediction is only considered correct if both the predicted class label is correct and the pose error is within a given threshold.

While this task has been studied in previous datasets [13, 24], ImageNet3D brings new challenges to existing models. Previous studies often focused on 12 or 20 categories [31, 32, 33] – how can we scale up these category-level 3D models to 200 categories while retaining a comparable performance? Moreover, with the meta classes and more categories available, we can better assess the limitations of current category-level pose estimation models.

**Task formulation.** For each of the 200 categories, we split the samples into training and validation splits, each accounting for about 50% of the data. Based on the number of samples used for training, we can further evaluate models under zero-shot, few-shot, and fully supervised settings.

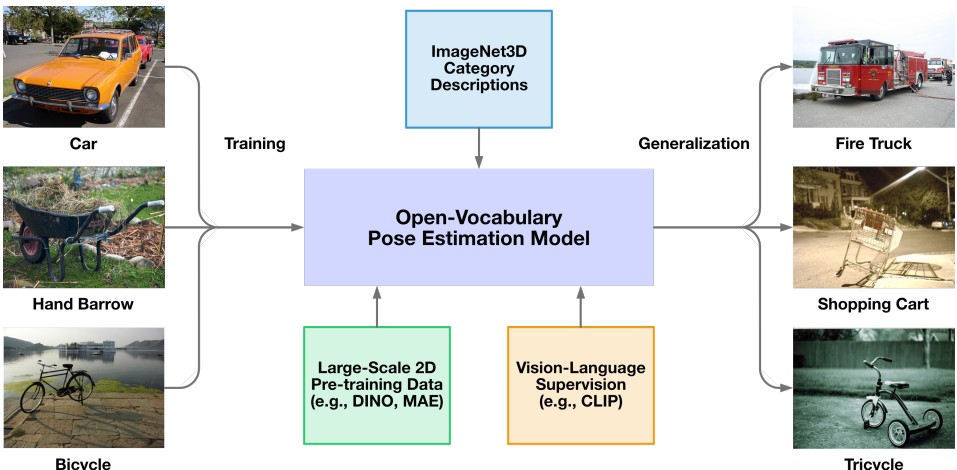

Figure 8: **Illustration of open vocabulary pose estimation.** Open-vocabulary models may utilize large-scale 2D data, vision-language supervision, or our category descriptions to learn transferable features and generalize to novel rigid categories.

**Evaluation.** Following [33], we adopt the **3D-aware classification accuracy**, where a prediction is correct only if the predicted class label is correct and the predicted pose error is lower than a given threshold.

## 5 Experimental Results

In this section we report the baseline performance of linear probing of object-level 3D awareness in Section 5.1, open-vocabulary pose estimation in Section 5.2, and joint image classification and category-level pose estimation in Section 5.3. For implementation details of various baseline models, including hyperparameters and hardware setup, please refer to Section B in the appendix. All experimental results in this section are based on the first version of ImageNet3D with 189 categories. Please refer to our dataset page for new releases of ImageNet3D and updated baseline results.

### 5.1 Linear Probing of Object-Level 3D Awareness

**Baselines.** We measure the object-level 3D awareness for a range of general-purpose vision models designed for representation learning [52, 49, 19, 51], multi-modal learning [20], and depth estimation [50]. These models adopt standard transformer architectures and we train a linear probe on frozen class embedding features. We focus on model sizes comparable to ViT-base and report the training supervisions and datasets in Table 2.

**Results.** We report the pose estimation accuracies with threshold $\pi/6$ for various baseline methods in Table 2. Results show that visual foundation models trained without 3D supervision demonstrates a reasonable level of object-level 3D awareness. Specifically, we find that DINO v2 largely outperforms other approaches in terms of object-level 3D awareness, followed by MAE, DINO, and MiDaS. However, the gap between these models are much smaller than the findings in [22]. Our ImageNet3D provides valuable assets to assess these visual foundation models from the perspective of object-level 3D awareness. In Section E.1 we present results on different metrics and study the scaling properties of self-supervised approaches on object-level 3D awareness.

### 5.2 Open-Vocabulary Pose Estimation

**Baselines.** Open-vocabulary pose estimation is a rather new topic, and there are no existing baselines designed specifically for this task. Oryon [34] operates on RGBD data and requires an image of the same object from a different viewpoint as a reference. OV9D [35] studies the problem by generating photorealistic synthetic data but the code is not available for reproduction. Hence for baseline results, we consider models that learn category-agnostic features that generalize to novel categories and

| Model | Arch. | Supervision | Dataset | Pose Acc@$\pi/6$ ↑ | | | | | | | |
|---|---|---|---|---|---|---|---|---|---|---|---|
| | | | | Avg. | Elec. | Fur. | Hou. | Mus. | Spo. | Veh. | Work |
| DeIT III [52] | ViT-B/16 | classification | ImageNet21k | 36.6 | 47.9 | 48.2 | 36.8 | 21.5 | 16.6 | 35.0 | 25.3 |
| MAE [49] | ViT-B/16 | SSL | ImageNet1k | 46.6 | 57.6 | 67.8 | 40.2 | 29.0 | 20.2 | 58.4 | 25.6 |
| DINO [19] | ViT-B/16 | SSL | ImageNet1k | 42.0 | 53.1 | 57.0 | 39.8 | 28.0 | 19.3 | 45.3 | 27.0 |
| DINO v2 [51] | ViT-B/14 | SSL | LVD-142M | 56.3 | 64.0 | 75.3 | 47.9 | 32.9 | 23.5 | 74.7 | 38.1 |
| CLIP [20] | ViT-B/16 | VLM | *private* | 39.7 | 50.3 | 52.8 | 39.7 | 23.1 | 19.3 | 39.8 | 26.4 |
| MiDaS [50] | ViT-L/16 | depth | MIX-6 | 40.5 | 50.9 | 56.7 | 40.2 | 26.7 | 18.9 | 39.2 | 28.1 |

Table 2: **Quantitative results on probing of object-level 3D awareness.** We report the $\pi/6$ *pose estimation accuracy* for the average performance on all categories, as well as the performance for each meta class (from left to right): *electronics*, *furniture*, *household items*, *music instrument*, *sports equipment*, *vehicles & transportation*, and *work equipment*. Among the tested visual foundation models, DINO v2 demonstrated the best object-level 3D awareness.

| Model | Novel Categories - Pose Acc@$\pi/6$ ↑ | | | | | | | |
|---|---|---|---|---|---|---|---|---|
| | Avg. | Electronics | Furniture | Household | Music | Sports | Vehicles | Work |
| ResNet50-General *(trained on novel categories)* | 53.6 | 49.2 | 52.4 | 45.8 | 26.0 | 65.2 | 56.5 | 58.5 |
| ResNet50-General | 37.1 | 30.1 | 35.6 | 28.1 | 11.8 | 51.7 | 36.7 | 40.9 |
| SwinTrans-T-General | 35.8 | 30.9 | 34.3 | 26.1 | 12.2 | 46.2 | 34.4 | 39.2 |
| NMM-Sphere | 29.5 | 31.7 | 25.4 | 21.7 | 25.6 | 19.8 | 33.4 | 19.3 |

Table 3: **Quantitative results on open-vocabulary pose estimation.** We report the *pose estimation accuracy* with threshold $\pi/6$ on testing data from novel categories unseen during training. We report the average performance on all novel categories, as well as performance for novel categories in each meta class. Results show that models with category-agnostic features can generalize to novel categories, but by a limited amount.

instances. Two types of approaches are considered: **(i) Classification-based methods** that formulate pose estimation as a classification problem. A pose classification head is trained on top of the backbone features. We consider two types of backbones, ResNet50 and SwinTransformer-Tiny, as our baselines. **(ii) 3D compositional models** learn neural mesh models with contrastive features and perform analysis-by-synthesis during inference. We develop *NMM-Sphere*, which is a 3D compositional model with a general sphere mesh for all categories and trained with class and part contrastive features [33].

**Results.** We report the pose estimation accuracy with threshold $\pi/6$ in Table 3 and present the full results in Section E.2. Results show that by annotating cross-category 3D alignment, models trained with category-agnostic features can generalize to novel categories with a reasonable performance. However, generalization abilities of current 3D models are still quite limited when compared to models trained on annotations from novel categories. Open-vocabulary pose estimation is a rather new topic but is crucial to the development of general-purpose 3D understanding. We call for future studies on this challenging but important problem.

## 5.3 Image Classification and Category-Level Pose Estimation

**Baselines.** We consider two types of baseline methods: **(i) Classification-based methods** that formulate pose estimation as a classification problem and train a shared pose classification head. Following previous works [31, 33], we extend ResNet50 and SwinTransformer-Tiny for pose estimation, denoted by "ResNet50-General" and "SwinTrans-T-General". **(ii) 3D compositional models** learn neural mesh models with contrastive features and perform analysis-by-synthesis during inference. NOVUM [33] adopts category-level meshes and more robust rendering techniques. We develop *NMM-Sphere*, which is a 3D compositional model with a general sphere mesh for all categories and is trained with class and part contrastive features [33].

**Results.** We report the 3D-aware classification accuracy with threshold $\pi/6$ in Table 4. Results show that with ImageNet3D, we can develop general-purpose models capable of inferring 3D information

| Model | 3D-Aware Acc@$\pi/6$ ↑ | | | | | | | |
|---|---|---|---|---|---|---|---|---|
| | Avg. | Electronics | Furniture | Household | Music | Sports | Vehicles | Work |
| ResNet50-General | 50.9 | 60.0 | 67.2 | 43.0 | 43.8 | 27.7 | 64.1 | 33.8 |
| SwinTrans-T-General | 53.2 | **63.1** | **71.6** | 44.8 | 45.3 | 30.4 | 66.2 | 35.0 |
| LLaVA-pose | 49.1 | 58.0 | 65.6 | 41.6 | 41.0 | 26.1 | 61.8 | 32.1 |
| NOVUM [33] | 56.2 | 59.6 | 65.6 | **52.5** | 41.9 | 30.6 | **69.6** | 39.3 |
| NMM-Sphere | **57.4** | 61.3 | 65.9 | 52.4 | **51.7** | **40.5** | 67.9 | **43.4** |

Table 4: **Quantitative results on joint image classification and category-level pose estimation.** We report the *3D-aware classification accuracy* with threshold $\pi/6$ for the average performance, as well as performance for each meta class. Results show that with ImageNet3D, we can develop unified 3D models capable of inferring 3D information for a wide range of rigid categories. However, we also identify limitations of current 3D models when scaling up to a lot more object categories.

for a wide range of common rigid categories. However, we also note that there is a clear performance degradation as the number of categories scale up, as compared to results found in previous works [33]. We present the full results with other metrics and backbones and study the scaling properties of pose estimation models in Section E.3.

# 6 Conclusion

In this paper we present ImageNet3D, a large dataset for general-purpose object-level 3D understanding. ImageNet3D largely extends the number of rigid categories and object instances, as compared to previous datasets with 3D annotations. Moreover, ImageNet3D improves the quality of 3D annotations by annotating cross-category 3D alignment, and provides new types of annotations, such as object visual qualities and image captions interleaved with 3D information that enable new research problems. We provide baseline results on standard 3D tasks, as well as novel tasks such as probing of object-level 3D awareness and open-vocabulary pose estimation. Experimental results show that with ImageNet3D, we can develop general-purpose models capable of inferring 3D information for a wide range of rigid categories. We also identify limitations of existing 3D models from our baseline experiments and discuss new problems and challenges for future studies.

**Limitations.** As our image data are collected from ImageNet21k, most images are object-centric with only one or two instances. Thus our dataset may not be suitable for 3D object detection or tasks that require object co-occurrences. As far as we know, there are no existing 3D object detection datasets with 3D annotations for more than 20 categories. One reason is that annotating 6D poses for multiple categories is very time consuming, and category co-occurrences follow a long-tail distribution. On the other hand, previous studies [32, 33] have found that compositional models trained on object-centric data have the ability to generalize to real images with multiple objects or partial occlusion, which makes our ImageNet3D a competitive option when developing models for general-purpose object-level 3D understanding.

# Acknowledgements

Alan Yuille acknowledges support from the Army Research Laboratory W911NF2320008 and National Eye Institute (NEI) with Award ID: R01EY037193. Adam Kortylewski acknowledges support for his Emmy Noether Research Group funded by the German Science Foundation (DFG) under Grant No. 468670075.

We would like to thank David Chauca, Olga Kuzmich, Yi Luo, Soh Kay Leen, Chenhao Lin, Hardik Shah, Jishuo Yang, Yuqi Li, Manqin Cai, Nanru Dai, Shen Wang, Wanyi Dai, Yifan Shuai, Zhangbo Cheng for their valuable help with the ImageNet3D data collection.

We are also grateful to the anonymous reviewers for the valuable discussions and constructive feedback.

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

## Appendix Contents

In the appendix, we provide the following materials:

## A  Details about ImageNet3D Construction

### A.1  Dataset Details

**Choice of rigid categories.** Our goal is to annotate most common rigid categories in natural images. We choose ImageNet21k as the source of our images as it includes a wide range of diverse images with object labels. Given the 21k categories in ImageNet21k, we choose the 200 categories based on the following criteria:

1. **The category must be rigid.** Categories such as animals, clothes, or food are skipped as they are not suitable for category-level pose estimation.
2. **The category must be a general class of objects, rather than a specific and limited subset of objects.** For instance, categories such as Gondola or speed boats are skipped as they are considered as subsets of boats.
3. **The category must have well-defined shapes.** Categories such as chimes vary too much in shapes, making it hard to define common canonical poses for object pose estimation.
4. **Objects from this category must vary in 3D viewpoints.** Images of oscilloscopes are often taken from the same viewpoint (*i.e.*, front), making this category trivial for pose estimation.

**Removing ambiguity in 3D viewpoint.** Objects from certain categories have ambiguities in terms of 3D viewpoint. For instance, tables when looked from "front" and "back" are visually indistinguishable. Certain datasets resolve this issue by annotating symmetry axes [35]. We follow [13, 24] and resolve the ambiguity by defining a "common" viewpoint. For instance, we assume tables are always looked at from the "front" and bottles always have zero azimuth. Models would learn such biases from the training data and we could adopt the standard pose estimation metrics during evaluation.

For other information, please refer to our datasheet for dataset.

### A.2  Annotator Guidelines

To improve the quality of the 3D annotations, we provide detailed guidelines and tutorials to the annotators. These documents provide a detailed introduction to each parameter to be annotated, how to use the web app, and how boundary cases should be handled. We refer the readers to Section C.3 where we provide links to our annotator tutorials and guidelines.

### A.3  Caption Generation

In ImageNet3D we provide two types of captions, natural image captions interleaved with 3D information 3.3 and category-level captions that provide detailed descriptions of object shape, part structure, and how humans interact with these objects for all categories in ImageNet3D.

**Natural image captions interleaved with 3D information.** We follow [6] and adopt a GPT-assisted approach to generate these captions. Specifically, we provide GPT-4v with our 3D annotations as context information and generate natural captions summarizing the objects in the images, as well

as the spatial and structural information. To generate captions interleaved with 3D information, we assign names (*e.g.*, CAR1 and BICYCLE1) to each object in the images, which are fed into GPT-4v along with each object's bounding box. Once we obtain the captions from GPT-4v, we insert 3D annotations after the first mention of the objects names, *i.e.*, from "CAR1" to "CAR1 with <pose6d>". Please refer to Figure 9 for examples of our system and user prompts.

**Category-level captions.** We manually annotate category-level captions that describe in details each category's object shape, part structure, and how humans interact with these objects for all categories in ImageNet3D.

### A.4 Ethics and Institutional Review Board (IRB)

We follow the ethics guidelines of NeurIPS and obtained Institutional Review Board (IRB) approvals prior to the start of our work. We described potential risks to the annotators, such as being exposed to inappropriate images from the ImageNet21k dataset [23], and explained the purpose of the study and how the collected data will be used. All annotators are paid by a fair amount as required at our institution. Link to our IRB approval: drive.google.com.

## B Implementation Details

### B.1 Baseline Models

**Classification-based methods.** Classification-based methods formulate pose estimation as a classification problem. Three linear classifiers are added on top of feature backbones with respect to the three pose parameters. Continuous values from 0 to $2\pi$ are projected into 40 bins, which are then learned with a cross-entropy loss by the classifier heads. All classification-based methods are trained on one A5000 GPUs for less than one day, depending on the backbone size.

**3D compositional models.** For NOVUM [33], we simply follow the official implementation. For NMM-Sphere, we extend a neural mesh model with a unified sphere mesh shared by all categories. The NMM-Sphere model could be applied for joint classification and pose estimation with a class-contrastive loss, or be used for open-vocabulary pose estimation by learning category-agnostic features. All 3D compositional models are trained on eight A5000 GPUs for about two days.

**LLaVA-pose.** Similar to [25, 26], we extend the LLaVA [6] model with a <pose> token, which is then decoded with a MLP (*i.e.*, a classifier head) to predict the pose. The LLaVA-pose model is trained on eight A5000 GPUs for one day.

### B.2 Training Details

**Data augmentations.** Our goal is to present baseline performance on ImageNet3D so we avoid complex data augmentations and leave it for future work to explore the benefits of data augmentation. For baseline models on all three tasks, we only adopt random horizontal flip.

**Linear probing of object-level 3D awareness.** Following [51] we grid search learning rates, pooling strategies, and backbone blocks (where features are taken from) and report the validation accuracy achieved by the best set of parameters.

**Open-vocabulary pose estimation.** For classification-based methods, models are trained for 120 epochs with a batch size of 64. We adopt the SGD classifier with an initial learning rate of 0.01.

**Joint image classification and category-level pose estimation.** We adopt the same training strategy as in open-vocabulary pose estimation. Moreover, for weights balancing the classification loss and the pose estimation loss, we simply choose $w_1 = w_2 = 1.0$.

### B.3 Data Pre-processing for Linear Probing

In the linear probing of object-level 3D awareness experiments, we train linear classifiers on top of frozen feature representation from various visual foundation models. To ensure that the neural features encode rich information about the target object, we adopt a data pre-processing step where we crop and resize the image based on the 2D and 3D annotations. In the processed images, objects

are centered in the 2D image plane and have a (roughly) consistent size. Qualitative examples are demonstrated in Figure 7.

## C    Data and Code Release

### C.1    License

Our ImageNet3D dataset, including 3D and other annotations, are released under the ATTRIBUTION-NONCOMMERCIAL 4.0 INTERNATIONAL license, *i.e.*, CC BY-NC 4.0. Additionally users should abide to the terms of access and license from the original ImageNet.

### C.2    Risks and Concerns

**Harmful contents.** A very few number of images in ImageNet3D may contain data that, if viewed directly, might be offensive, insulting, threatening, or might otherwise cause anxiety. These images are taken directly from ImageNet21k so please follow the guidelines of ImageNet21k.

**Personally identifiable information.** Certain images may contain faces to identify individuals. However, these images are taken directly from ImageNet21k so please follow the guidelines of ImageNet21k.

### C.3    ImageNet3D Dataset and Code

1. **Datasheet for dataset:** github.com
2. **Raw data:** huggingface.co
3. **Croissant metadata:** huggingface.co
4. **Source code for main experiments:** github.com
5. **Source code of our web app:** github.com
6. **Annotator tutorial:** drive.google.com
7. **Annotator guidelines:** drive.google.com

## D    Human Evaluation of Annotation Quality

As there is an overlap of images between ObjectNet3D [24] and ImageNet3D, we analyze the quality of the annotations with human evaluation. Specifically, we present the annotated 6D poses from ObjectNet3D and ImageNet3D side by side to human evaluators. Then the human evaluator must choose which annotation is correct and visually better. The evaluation metrics include both the alignment of 3D viewpoint, as well as the 3D location of the object. We randomly shuffle the order the annotations presented to the annotators.

We collect human evaluation results from 16 categories that are both annotated in ImageNet3D and ObjectNet3D. 50 images are sampled from each category to compare the annotation quality. Results show that for all categories tested, annotations from ImageNet3D are generally favored than the annotations from ObjectNet3D, and on average, for 73.25% of the samples, annotations from ImageNet3D are favored. This demonstrate that 3D annotations from ImageNet3D tend to have a better quality than the annotations in ObjectNet3D. Detailed results are presented in Table 5.

## E    Additional Experimental Results

### E.1    Linear Probing of Object-Level 3D Awareness

We report baseline performance using both $\pi/6$ and $\pi/18$ pose accuracies in Table 6. Moreover, we study the scaling properties of various baseline methods and visualize the results in Figure 10. Results show that DINO v2 and DeiT are not scaling well as model parameters increase, and MAE outperforms DINO v2 in large and huge model sizes.

## E.2 Open-Vocabulary Pose Estimation

We report baseline performance using both $\pi/6$ and $\pi/18$ pose accuracies in Table 7.

## E.3 Joint Image Classification and Category-Level Pose Estimation

We report baseline performance using $\pi/6$ and $\pi/18$ pose accuracies, as well as the median pose error, in Table 7. Furthermore, we show in Figure 11 the scaling properties of ResNet and SwinTransformer on joint image classification and category-level pose estimation. Results show that SwinTransformer obtains better results when model sizes are comparable to ViT-B but ResNet outperforms Swin Transformer as the model sizes scale up.

## E.4 Ablation Study on Cross-Category 3D Alignment

In ImageNet3D we adopt cross-category 3D alignment by aligning the canonical poses for all object categories. This design resolves the ambiguity of canonical poses in novel object pose estimation. Moreover, we find that with cross-category 3D alignment, models can learn shared semantics between different categories, yielding a higher benchmark performance. Specifically we compare the benchmark performance before and after a random rotation (multiple of 90 degrees) is added to the canonical poses of about 1/3 of the object categories. We report the quantitative results in Table 9. The results highlight the benefits of joint training on cross-category aligned data when developing unified 3D vision models.

Note that NMM-Sphere fails to converge well without cross-category 3D alignment. The reason is that misaligned canonical poses lead to false negative pairs and break the part contrastive learning. For instance, if "shoe" and "slipper" have misaligned canonical pose, then the shared semantic parts would form negative pairs in contrastive learning. Instead, shared semantic parts would form positive pairs and produce similar part embeddings when cross-category 3D alignment is adopted.

## E.5 Ablation Study on Training Time

We analyze the computational costs of training joint classification and pose estimation models on our ImageNet3D with 200 categories and on a subset of ImageNet3D with 100 categories (same categories as ObjectNet3D). We visualize the performance by wall clock time in Figure 12 of the attached PDF.

Results show that for CNN or transformer models that formulate pose estimation as a classification problem (*i.e.*, ResNet or SwinTransformer), they generally converge pretty fast and take a similar amount of time to converge on ImageNet3D or the 100-class subset. These methods don't require many computational resources to train.

Meanwhile neural mesh models (*i.e.*, NMM-sphere) take much longer to train and converge. This is because these models learn part-contrastive features with contrastive losses, which limit the training batch size and generally don't converge as fast as standard classification objectives. For future work we will explore methods to improve the training efficiency of neural mesh models, such as by involving better foreground/background feature sampling methods or by mining hard-negative pairs.

**Image caption generation with GPT-4v**

**System:**
You are an AI visual assistant that can analyze a single image. You receive an image and a list of main objects, with the object names in the parentheses and object locations in the image given by detailed coordinates. These coordinates are in the form of bounding boxes, represented as (x1, y1, x2, y2) with floating numbers ranging from 0 to 1. These values correspond to the top left x, top left y, bottom right x, and bottom right y.

When describing the given objects, mention the names of the objects as their reference. Include details like object counts, position of the objects, relative position between the objects.

**User:**
Describe the image in details. There are 2 known objects: BICYCLE1 with bounding box [...], and BICYCLE2 with bounding box [...]
Mention the names of the objects as their reference.

Figure 9: **GPT-assisted approach to generate image caption interleaved with 3D information.** Following [6], we provide our 3D annotations and object names as context information to GPT-4v and generate natural image captions interleaved with 3D information. Such captions can be used to integrate general-purpose 3D models with large language models [25, 26].

| Category | Ours favored | Category | Ours favored | Category | Ours favored |
|---|---|---|---|---|---|
| computer | 90% | helmet | 56% | coffee maker | 66% |
| mouse | 70% | fire extinguisher | 66% | backpack | 94% |
| boat | 88% | train | 64% | piano | 76% |
| bicycle | 94% | teapot | 56% | suitcase | 85% |
| calculator | 74% | flashlight | 62% | watch | 72% |
| bucket | 60% | | | | |

Table 5: **Human evaluation of 3D annotations between ImageNet3D and ObjectNet3D.** For each of the 16 categories, we sample 50 images and present to the annotators to compare the annotation qualities. In this table, we present the percentage of samples where the 3D annotation from ImageNet3D is considered better than the 3D annotation from ObjectNet3D.

| Model | Arch. | Supervision | Dataset | Pose Acc@$\pi/6$ | | | | | | | |
|---|---|---|---|---|---|---|---|---|---|---|---|
| | | | | Avg. | Elec. | Fur. | Hou. | Mus. | Spo. | Veh. | Work |
| DeIT III [52] | ViT-B/16 | classification | ImageNet21k | 36.6 | 47.9 | 48.2 | 36.8 | 21.5 | 16.6 | 35.0 | 25.3 |
| MAE [49] | ViT-B/16 | SSL | ImageNet1k | 46.6 | 57.6 | 67.8 | 40.2 | 29.0 | 20.2 | 58.4 | 25.6 |
| DINO [19] | ViT-B/16 | SSL | ImageNet1k | 42.0 | 53.1 | 57.0 | 39.8 | 28.0 | 19.3 | 45.3 | 27.0 |
| DINO v2 [51] | ViT-B/14 | SSL | LVD-142M | **56.3** | **64.0** | **75.3** | **47.9** | **32.9** | **23.5** | **74.7** | **38.1** |
| CLIP [20] | ViT-B/16 | VLM | *private* | 39.7 | 50.3 | 52.8 | 39.7 | 23.1 | 19.3 | 39.8 | 26.4 |
| MiDaS [50] | ViT-L/16 | depth | MIX-6 | 40.5 | 50.9 | 56.7 | 40.2 | 26.7 | 18.9 | 39.2 | 28.1 |

| Model | Arch. | Supervision | Dataset | Pose Acc@$\pi/18$ | | | | | | | |
|---|---|---|---|---|---|---|---|---|---|---|---|
| | | | | Avg. | Elec. | Fur. | Hou. | Mus. | Spo. | Veh. | Work |
| DeIT III [52] | ViT-B/16 | classification | ImageNet21k | 14.4 | 19.1 | 20.4 | 14.0 | 7.1 | 5.9 | 13.4 | 11.0 |
| MAE [49] | ViT-B/16 | SSL | ImageNet1k | 21.7 | 26.4 | 35.5 | 18.1 | 10.5 | 7.7 | 27.1 | 11.9 |
| DINO [19] | ViT-B/16 | SSL | ImageNet1k | 18.7 | 23.2 | 28.7 | 16.5 | 9.3 | 8.4 | 20.8 | 12.3 |
| DINO v2 [51] | ViT-B/14 | SSL | LVD-142M | **26.1** | **28.4** | **40.6** | **20.6** | **12.0** | **9.7** | **36.3** | **17.0** |
| CLIP [20] | ViT-B/16 | VLM | *private* | 16.8 | 21.2 | 25.0 | 16.0 | 7.5 | 6.2 | 17.2 | 11.6 |
| MiDaS [50] | ViT-L/16 | depth | MIX-6 | 17.4 | 22.1 | 26.7 | 16.7 | 8.4 | 8.1 | 16.4 | 12.5 |

Table 6: **Quantitative results on probing of object-level 3D awareness.** We report the $\pi/6$ *pose estimation accuracy* for the average performance on all categories, as well as the performance for each meta class (from left to right): *electronics*, *furniture*, *household items*, *music instrument*, *sports equipment*, *vehicles & transportation*, and *work equipment*. Among the tested visual foundation models, DINO v2 demonstrated the best object-level 3D awareness.

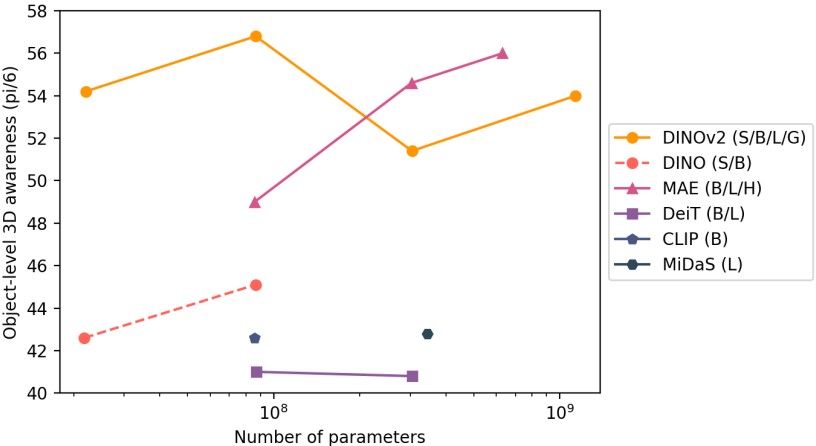

Figure 10: **Scaling properties of various backbones on linear probing of object-level 3D awareness.**

| Model | Novel Categories - Pose Acc@$\pi/6$ ↑ | | | | | | | |
|---|---|---|---|---|---|---|---|---|
| | Avg. | Electronics | Furniture | Household | Music | Sports | Vehicles | Work |
| ResNet50-General *(trained on novel categories)* | 53.6 | 49.2 | 52.4 | 45.8 | 26.0 | 65.2 | 56.5 | 58.5 |
| ResNet50-General | **37.1** | 30.1 | **35.6** | **28.1** | 11.8 | **51.7** | **36.7** | **40.9** |
| SwinTrans-T-General | 35.8 | 30.9 | 34.3 | 26.1 | 12.2 | 46.2 | 34.4 | 39.2 |
| NMM-Sphere | 29.5 | **31.7** | 25.4 | 21.7 | **25.6** | 19.8 | 33.4 | 19.3 |

| Model | Novel Categories - Pose Acc@$\pi/18$ ↑ | | | | | | | |
|---|---|---|---|---|---|---|---|---|
| | Avg. | Electronics | Furniture | Household | Music | Sports | Vehicles | Work |
| ResNet50-General *(trained on novel categories)* | 25.5 | 25.9 | 23.3 | 19.2 | 11.8 | 31.0 | 27.4 | 28.2 |
| ResNet50-General | **13.5** | **13.2** | 12.4 | **9.0** | 2.1 | **21.8** | **13.1** | **15.0** |
| SwinTrans-T-General | 13.1 | **13.2** | **12.7** | 8.1 | 1.7 | 18.0 | 11.9 | 13.6 |
| NMM-Sphere | 6.0 | 6.6 | 4.4 | 3.5 | **3.1** | 4.7 | 6.2 | 2.8 |

Table 7: **Quantitative results on open-vocabulary pose estimation.** We report the *pose estimation accuracy* with threshold $\pi/6$ on testing data from novel categories unseen during training. We report the average performance on all novel categories, as well as performance for novel categories in each meta class. Results show that models with category-agnostic features can generalize to novel categories, but by a limited amount.

| Model | 3D-Aware Acc@$\pi/6$ ↑ | | | | | | | |
|---|---|---|---|---|---|---|---|---|
| | Avg. | Electronics | Furniture | Household | Music | Sports | Vehicles | Work |
| ResNet50-General | 50.9 | 60.0 | 67.2 | 43.0 | 43.8 | 27.7 | 64.1 | 33.8 |
| SwinTrans-T-General | 53.2 | **63.1** | **71.6** | 44.8 | 45.3 | 30.4 | 66.2 | 35.0 |
| LLaVA-pose | 49.1 | 58.0 | 65.6 | 41.6 | 41.0 | 26.1 | 61.8 | 32.1 |
| NOVUM [33] | 56.2 | 59.6 | 65.6 | **52.5** | 41.9 | 30.6 | **69.6** | 39.3 |
| NMM-Sphere | **57.4** | 61.3 | 65.9 | 52.4 | **51.7** | **40.5** | 67.9 | **43.4** |

| Model | 3D-Aware Acc@$\pi/18$ ↑ | | | | | | | |
|---|---|---|---|---|---|---|---|---|
| | Avg. | Electronics | Furniture | Household | Music | Sports | Vehicles | Work |
| ResNet50-General | 25.3 | 28.6 | 35.5 | 19.0 | 16.3 | 13.2 | 36.1 | 16.2 |
| SwinTrans-T-General | **27.4** | **31.2** | **40.1** | 19.9 | **19.3** | **15.4** | **39.2** | **16.9** |
| LLaVA-pose | 15.2 | 16.2 | 20.5 | 11.7 | 10.8 | 9.0 | 23.0 | 9.2 |
| NOVUM [33] | 21.7 | 22.5 | 26.2 | 18.4 | 10.5 | 6.6 | 33.8 | 9.5 |
| NMM-Sphere | 22.8 | 23.2 | 31.4 | **20.1** | 14.5 | 10.7 | 32.7 | 9.2 |

| Model | Median Pose Error ↓ | | | | | | | |
|---|---|---|---|---|---|---|---|---|
| | Avg. | Electronics | Furniture | Household | Music | Sports | Vehicles | Work |
| ResNet50-General | 28.6 | 19.6 | 14.5 | 46.9 | 38.0 | 88.5 | 16.3 | 67.5 |
| SwinTrans-T-General | 25.6 | **17.1** | **12.4** | 40.8 | 35.4 | 68.0 | **14.7** | 64.5 |
| LLaVA-pose | 31.2 | 22.5 | 17.3 | 49.5 | 40.7 | 90.3 | 19.1 | 70.1 |
| NOVUM [33] | 24.4 | 22.1 | 18.6 | **27.6** | 35.7 | 58.1 | 16.0 | 41.6 |
| NMM-Sphere | **23.7** | 21.4 | 17.4 | **27.6** | **28.7** | **43.4** | 17.0 | **36.2** |

Table 8: **Quantitative results on joint image classification and category-level pose estimation.** We report the *3D-aware classification accuracy* with threshold $\pi/6$ for the average performance, as well as performance for each meta class. Results show that with ImageNet3D, we can develop unified 3D models capable of inferring 3D information for a wide range of rigid categories. However, we also identify limitations of current 3D models when scaling up to a lot more object categories.

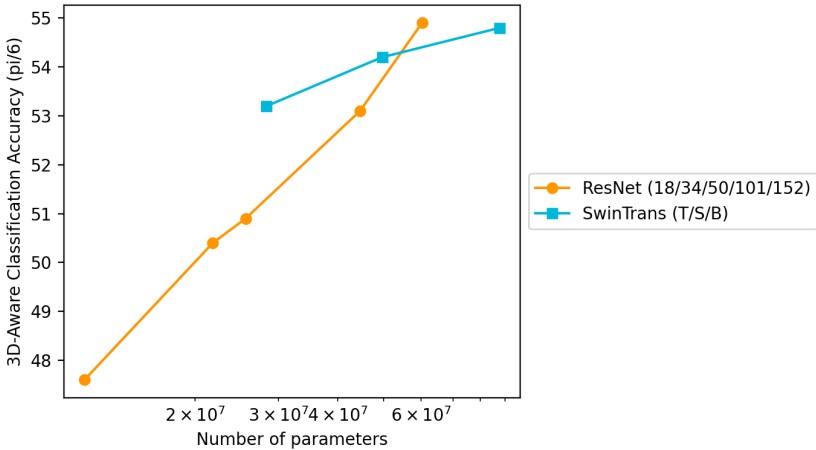

Figure 11: **Scaling properties of ResNet-50 and Swin Transformer on joint image classification and category-level pose estimation.**

| Model | w/ 3D Alignment | Novel Categories - Pose Acc@$\pi/6$ ↑ | | | | | | | |
|---|---|---|---|---|---|---|---|---|---|
| | | Avg. | Electronics | Furniture | Household | Music | Sports | Vehicles | Work |
| ResNet50-General | no | 47.6 | 56.9 | 63.0 | 40.0 | 39.0 | 27.1 | 59.3 | 32.0 |
| ResNet50-General | yes | 50.9 | 60.0 | 67.2 | 43.0 | 43.8 | 27.7 | 64.1 | 33.8 |
| SwinTrans-T-General | no | 49.8 | 60.0 | 67.0 | 42.2 | 43.6 | 29.6 | 60.5 | 32.6 |
| SwinTrans-T-General | yes | 53.2 | 63.1 | 71.6 | 44.8 | 45.3 | 30.4 | 66.2 | 35.0 |
| NMM-Sphere | no | 10.6 | 3.8 | 12.1 | 6.7 | 1.2 | 2.1 | 26.0 | 4.2 |
| NMM-Sphere | yes | 57.4 | 61.3 | 65.9 | 52.4 | 51.7 | 40.5 | 67.9 | 43.4 |

Table 9: **Ablation study on the benefits of joint training on cross-category aligned data.** Results show that with cross-category 3D alignment, models can learn shared semantics between different categories, yielding a higher benchmark performance. This highlights the benefits of joint training on cross-category aligned data when developing unified 3D vision models.

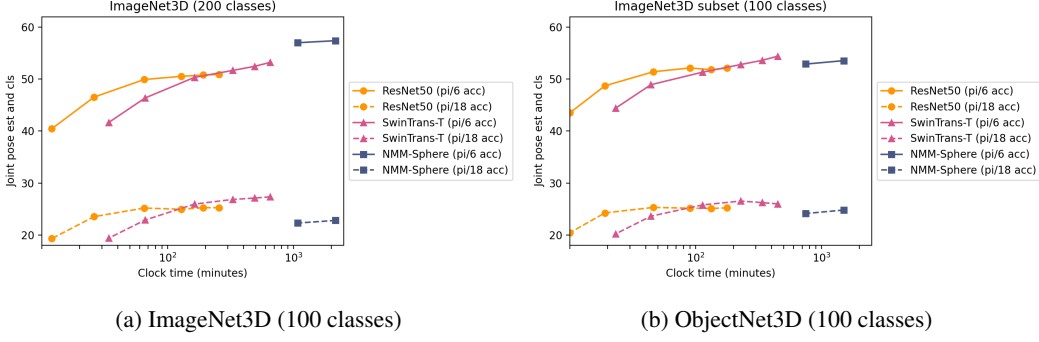

(a) ImageNet3D (100 classes)                                   (b) ObjectNet3D (100 classes)

Figure 12: Models' performance by wall clock time on ImageNet3D with 200 classes (left) and ObjectNet3D with 100 classes (right).

