# OpenReview forum: "ImageNet3D: Towards General-Purpose Object-Level 3D Understanding"
_NeurIPS.cc/2024/Datasets_and_Benchmarks_Track — NeurIPS 2024 Track Datasets and Benchmarks Poster_

### Official Review · Reviewer_pu6E · 2024-07-21
**This work presents a large dataset for general-purpose object-level 3D understanding  and defines two new tasks.**

**Rating:** 6
**Confidence:** 4
**Correctness:** Correct
**Clarity:** yes

**Review:**

This paper is clearly expressed with a logical structure, demonstrating a certain influence.
Pros：
1.ImageNet3D vastly broadens object categories (200) and instances (86k+), with rich 3D annotations, outstripping prior datasets in size and variety. 2.This dataset encompasses standard 3D annotations plus cross-category alignments & language captions integrated with 3D data, boosting multi-modal studies.
3.The dataset's distinct traits enable tackling novel research hurdles like object-level 3D comprehension & open-vocabulary pose prediction, advancing 3D object understanding. 4.Experiments show ImageNet3D-trained models accurately infer 3D info across diverse rigid objects, underscoring models' versatility & broad usage potential. Cons：
Considering the dataset's size and task complexity, computational efficiency is key. The paper should delve deeper into ImageNet3D's training requirements and suggest ways to reduce training time while maintaining high performance.

**Strengths:**

1. The introduction of ImageNet3D significantly expands the scope of object categories (200 categories) and instances (over 86,000 objects) with detailed 3D annotations, surpassing previous datasets in scale and diversity.
2. The dataset includes not just standard 3D annotations but also cross-category 3D alignments and natural language captions interleaved with 3D data, facilitating further multi-modal research.
3. The dataset and its unique features enable the exploration of new research challenges, such as probing object-level 3D awareness and open-vocabulary pose estimation, potentially driving forward the field of 3D object understanding.
4. The experimental results demonstrate that the models developed using ImageNet3D can effectively infer 3D information across a wide array of rigid object categories, showcasing the versatility and broad applicability of the models trained on this dataset.

**Additional Feedback:**

None

**Documentation:**

Complete

**Limitations:**

the authors have included the limitations

**Opportunities For Improvement:**

Given the large scale of the dataset and the complexity of the tasks, computational efficiency is a critical aspect. The paper could discuss more about the computational demands of training models on ImageNet3D and propose methods to optimize training time without sacrificing performance.

**Relation To Prior Work:**

Compared to previous datasets, it is innovative and presents new tasks.

**Summary And Contributions:**

The paper introduces ImageNet3D, a significant expansion of the traditional ImageNet dataset enhanced with 2D and 3D annotations for general-purpose object-level 3D understanding in vision models. It is designed to address the limitations of existing datasets that lack diversity in annotations, which restricts the generalization capabilities of models across unseen object categories. The contributions of this work include the development of ImageNet3D to enable comprehensive analysis of object-level 3D awareness, fostering the creation of versatile models capable of interpreting both 2D and 3D data from natural images, and enhancing these models' integration with large language models for advanced 3D reasoning. This work also introduces new tasks such as probing object-level 3D awareness and open vocabulary pose estimation to further test and refine the capabilities of vision models using ImageNet3D, demonstrating its potential to significantly improve the general-purpose 3D understanding of vision models.

---

> ### Author Rebuttal · Authors · 2024-08-17
>
> **`[Weakness 1]` Computational efficiency is a critical aspect. The paper should delve deeper into ImageNet3D's training requirements and suggest ways to reduce training time while maintaining high performance.**
>
> We thank the reviewer for the comment. Please refer to `[General Response 3]` where we present experimental results studying the computational costs of pose estimation models as dataset size scales.
>
> Specifically, we analyze the computational costs of training joint classification and pose estimation models on our ImageNet3D with 200 categories and on a subset of ImageNet3D with 100 categories (same categories as ObjectNet3D). We visualize the performance by clock time in `Figure R.4` of the attached PDF in General Response.
>
> Results show that for CNN or transformer models that formulate pose estimation as a classification problem (i.e., ResNet or SwinTransformer), they generally converge pretty fast and take a similar amount of time to converge on ImageNet3D or the 100-class subset. These methods don't require many computational resources to train.
>
> Meanwhile neural mesh models (i.e., NMM-sphere) take much longer to train and converge. This is because these models learn part-contrastive features with contrastive losses, which limit the training batch size and generally don’t converge as fast as standard classification objectives. For future work we will explore methods to improve the training efficiency of neural mesh models, such as by involving better foreground/background feature sampling methods or by mining hard-negative pairs.

---

### Official Review · Reviewer_ZDJV · 2024-07-23

**Rating:** 6
**Confidence:** 4
**Correctness:** Yes.
**Clarity:** Yes.

**Review:**

See strength and weakness.

**Strengths:**

- More types of annotations: the dataset has annotated 6D pose, captions, object
visual quality, cross-category, 3D alignment, which is significantly more than a pose estimation dataset.
- New benchmark tasks: The paper proposes open-vocabulary pose estimation, which could stimulate new research directions, and show be acknowledged.

**Additional Feedback:**

N/A

**Documentation:**

Code is provided. License is discussed in the supplemental. A maintainence plan and website are missing. In general, it looks fine to me.

**Ethics:**

No.

**Limitations:**

Yes, in the supplemental.

**Opportunities For Improvement:**

- Dataset size and significance: While ImageNet3D is larger than some existing datasets, the increase in size is not substantial given the time elapsed since previous works. Compared to ObjectNet3D (2016) with 100 categories and 57,000 instances, ImageNet3D's 200 categories and 86,000 instances represent a modest increase over an 8-year period. This raises questions about whether the dataset size is sufficient to significantly advance the field or help the research community.
- Experimental results and novelty: The paper's experimental findings, particularly in the linear probing of object-level 3D awareness, largely follow the approach of Banani et al. without presenting significant novel insights. This lack of groundbreaking results diminishes the overall impact and novelty of the work.
- Use case clarity: While the paper mentions potential applications in 3D-LLM integration, these use cases are not thoroughly explored or demonstrated in the experiments. This leaves uncertainty about the practical applications and benefits of the dataset in real-world scenarios.

**Relation To Prior Work:**

Yes.

**Summary And Contributions:**

This paper introduces ImageNet3D, a large-scale dataset aimed at advancing general-purpose object-level 3D understanding. The key contributions are:
- A dataset of 200 object categories from ImageNet, annotated with 2D bounding boxes, 6D poses, and 3D locations for over 86,000 objects.
- Cross-category 3D alignment annotations to enable better generalization across categories.
- Image captions interleaved with 3D information to facilitate integration with large language models.
- New benchmark tasks including probing object-level 3D awareness and open-vocabulary pose estimation.
- Baseline results on these tasks using various model architectures.

The authors argue that ImageNet3D addresses limitations of existing 3D datasets in terms of category diversity and annotation quality, enabling research on more general 3D understanding models. They present experimental results showing the potential of the dataset for developing models capable of inferring 3D information across a wide range of rigid object categories.

---

> ### Author Rebuttal · Authors · 2024-08-17
>
> **`[Weakness 1]` Increase in size not substantial when compared to ObjectNet3D published in 2016.**
>
> We thank the reviewer for the comment. However, we would like to elaborate our contribution from the following aspects:
> 1. **Advancing 3D vision research.** We argue that the impact and significance of our dataset is not determined solely by the number of object instances, but more importantly by the research problems enabled by our dataset that will advance 3D vision research. Rather than studying the same problem, i.e., classification and 3D pose estimation, on more object categories, we propose new tasks and research questions that are crucial to robotics and vision-language research which have received rising interests in recent years. Specifically, only with ImageNet3D, we can (i) comprehensively evaluate the 3D object-level awareness of visual foundation models as ImageNet3D covers most of the common rigid categories in real life, (ii) study how 3D vision models can generalize to unseen categories with semantically similar parts which was not possible given the limited number of categories in ObjectNet3D (see Fig. 5), and (iii) develop unified 3D models that can address challenging visual question answering that require 3D awareness (see L36-45). We believe these new research problems will enable research on unified 3D vision models that benefit downstream applications in robotics and vision-language (see L29-35).
> 2. **Efforts to build ImageNet3D.** Although ImageNet3D is not growing in size by a large factor, the efforts to build this dataset is significant. As supported by some other reviewers, previous 3D annotations in ObjectNet3D demonstrate limited quality, which restrained their use cases and impact. In ImageNet3D, a lot of efforts have been paid to refine existing 3D annotations, annotate new categories and objects, and provide diverse types of new annotations. More than 1,400 hours of manpower have been spent on this process and this number is still growing as we keep reviewing the annotation quality and fixing bad annotations.
> 3. **Why 200 categories.** The reason we stop at 200 categories is that 200 is a rough number of rigid categories common in real life that are suitable for pose estimation. Note that we focus on “categories” rather than finer “sub-categories”. For instance, it’s possible to involve more categories such as motorboats, barber chairs, boots, which we consider as sub-categories of some of our current 200 categories and choose to disregard.
>
> &nbsp;
>
> **`[Weakness 2]` The paper's experimental findings, particularly in the linear probing of object-level 3D awareness, largely follow the approach of Banani et al. without presenting significant novel insights. This lack of groundbreaking results diminishes the overall impact and novelty of the work.**
>
> We thank the reviewer for the comment. However, we would like to make the following arguments:
> 1. **NeurIPS policy.** According to the submission policy of NeurIPS 2024, “papers appearing less than two months before the submission deadline are generally considered concurrent to NeurIPS submissions.” We were aware of Banani et al. when we submitted our work to NeurIPS but our linear probing experiments were completed well before their paper appeared on arXiv in mid April.
> 2. **Overall impact.** Linear probing of object-level 3D awareness is only one of the three experimental settings considered in our work. Despite the widespread attention on visual foundation models, the other two experimental settings focused on novel and foundational problems in the 3D vision community. **We don’t think the results presented in Banani et al. would diminish the contribution and impact of our dataset.**
> 3. **Research focus.** Although both Banani et al. and our work studies the 3D knowledge of visual foundation models, we focus on very different aspects of 3D awareness, as supported by the data we adopted, i.e., depth, surface normal, and multi-view images v.s. images with 3D pose annotations. Specifically, results in Banani et al. will help the AIGC community to study how visual foundation models encode shape and surface normals that are crucial for 2D and 3D generative tasks that require 3D consistencies. Meanwhile, our work studies the **object-level 3D awareness** of visual foundation models, which would benefit the vision-language community in finding a strong visual encoder to address hard VQA questions that require 3D object understanding. **These findings would not be possible without our ImageNet3D dataset.**
> 4. **Key findings.** Our linear probing experiments produces novel and important findings and results that were not presented in Banani et al.: (i) Although CLIP falls far behind DINOv2 on low-level 3D tasks (e.g., about 40% gap on depth recall shown in Banani et al.), the gap between CLIP and DINOv2 is much small for object-level 3D awareness (40% and 56%), which shows that CLIP is still a competitive choice when building VLM to address 3D-related questions. (ii) Although MiDaS performs well for low-level tasks considered in Banani et al., MiDaS achieves comparable performance to CLIP in terms of object-level 3D awareness. This shows that when building 3D-aware VLMs, depth pertaining will not directly benefit high-level 3D recognition tasks.
> 5. **Technical aspects.** Linear probing experiments in Banani et al. largely follow the linear probing experiments on ImageNet, which have been widely studied and adopted in the community. Adopting linear probing to study 3D problems does not necessarily mean that we are heavily built on Banani et al.

---

> > ### Author Rebuttal · Authors · 2024-08-17
> >
> > **`[Weakness 3]` Potential applications in 3D-LLM integration not demonstrated in the experiments.**
> >
> > Following the designs in [25,26], we adopted a simplified setting where we finetune a LLaVA model that predicts the category and pose in the format of “Q: what is the category and pose of the object in the image? A: The object is a xxx with <pose>.” This is considered a baseline model for “joint classification and pose estimation” and the performance is reported in Table 4 denoted by “LLaVA-pose”. Results show that LLaVA-pose falls behind other pose estimation models by a small margin. However, this experiment shows that we can successfully integrate 3D object-level recognition in vision-language models.
> >
> > In this simplified version we did not finetune the LLaVA-pose model on natural image captions. Although captions interleaved with 2D/3D tokens enable a new interface for vision-language models to interact with human pose [25], segmentation maps [26], and object 6D pose as in our work, this is a fairly new problem and evaluation of natural captions with 2D/3D tokens or vision-language models with 2D/3D tokens is an active research problem. Prior work [25] only evaluated on standard pose estimation and failed to show that training on their data would lead to models with improved 3D understanding (rather than simply a better pose estimation model). Many concerns are left unaddressed, e.g., can these 3D-VLMs answer 3D-related questions better with explicit 3D representations? Hence fairly estimating the quality of the captions is a challenging research problem. For future work we hope to build stronger vision-language models with better 3D awareness and present promising evaluations to support these findings.

---

> > ### Comment · Reviewer_ZDJV · 2024-08-29
> >
> > Thanks for your response.
> >
> > > only with ImageNet3D, we can (i) comprehensively evaluate the 3D object-level awareness of visual foundation models as ImageNet3D covers most of the common rigid categories in real life
> >
> > I'm relatively neutral to this. I'm not fully convinced how important 3D object-level awareness is for vision foundation models.
> >
> > > NeurIPS policy. According to the submission policy of NeurIPS 2024, “papers appearing less than two months before the submission deadline are generally considered concurrent to NeurIPS submissions.” We were aware of Banani et al. when we submitted our work to NeurIPS but our linear probing experiments were completed well before their paper appeared on arXiv in mid April.
> >
> > I think it's important to point out Banani et al is concurrent work in the paper. Upon reading the paper, I felt the paper is a followup of Banani et al. But yes that resolves my weakness 2 completely. I think you should also revise your paper to clarify Banani et al is a concurrent work.
> >
> > > Following the designs in [25,26], we adopted a simplified setting where we finetune a LLaVA model that predicts the category and pose in the format of “Q: what is the category and pose of the object in the image? A: The object is a xxx with <pose>.”
> >
> > That's a good experiment I miss. Thanks for pointing it out.
> >
> > After reading the rebuttal, my weakness 2 and 3 are resolved. I'll raise my rating to 6.

---

> > ### Author Response · Authors · 2024-08-29
> > **Official Comment by Authors**
> >
> > Dear Reviewer ZDJV,
> >
> > We are glad to learn that our rebuttal addresses some of your concerns.
> >
> > &nbsp;
> >
> > **Regarding the importance of object-level 3D awareness for visual foundation models:** Our hypothesis is that having implicit 3D awareness or explicit 3D latent representation will not only empower visual foundation models for more applications in 3D vision, but also allow these vision models to recognize objects and concepts in the 3D world and in the long run, achieve human-like intelligence. Specifically there are rising interests in challenging current VLMs for spatial reasoning [A,B] and exploring VLMs for robotics planning and manipulation [C].
> >
> > However, we agree with the reviewer that these are active research questions and the benefit of 3D awareness is debatable. Existing visual foundation models have demonstrated impressive understanding of images given only 2D supervision and instruction-tuning data. We hope to delve more into the 3D awareness of visual foundation models and with the ImageNet3D dataset, explore how 3D data can benefit visual foundation models on various 2D and 3D tasks.
> >
> > &nbsp;
> >
> > Thanks again for your time and constructive suggestions. Please let us know if there are any further questions.
> >
> > &nbsp;
> >
> > [A] Cheng et al. SpatialRGPT: Grounded Spatial Reasoning in Vision-Language Models.
> >
> > [B] Chen et al. SpatialVLM: Endowing Vision-Language Models with Spatial Reasoning Capabilities.
> >
> > [C] Yang et al. Octopus: Embodied Vision-Language Programmer from Environmental Feedback.

---

### Official Review · Reviewer_rBvX · 2024-07-23
**Great work, but paper presentation should be improved.**

**Rating:** 7
**Confidence:** 5

**Review:**

The proposed dataset seems useful for various 3D tasks especially for 3D-LLM work.
However, their paper presentation makes me quite confusing while reading the paper.
For more comments, please follow the strengths and opportunities of improvements written below.

**Strengths:**

The paper identifies and addresses significant shortcomings in current 3D object annotation datasets, particularly their limited number of categories and poor annotation quality. To bridge this gap, the authors introduce ImageNet3D, an extensive and enhanced dataset built upon the renowned ImageNet. By incorporating 200 categories, ImageNet3D significantly increases the diversity and utility of available data. Furthermore, ImageNet3D offers detailed annotations, including 2D bounding boxes, 3D poses, 3D locations, and image captions enriched with 3D information. These comprehensive annotations enable a wide range of analyses and facilitate the development of robust models.

**Additional Feedback:**

With additional experiments and paper revision, I suggest this paper to be accepted.

**Clarity:**

Not really.
As written above, the entire pipeline is quite complicated. Thus, authors should provide a diagram that describes the entire pipeline.
Moreover, the paper lacks details for experiments.
For those who are familiar in this area could read this paper easily; however, non-experts should be confused about the contexts.
A website for this work should facilitate the understanding.

**Correctness:**

Authors noted one of the advantages of using this dataset is its canonically oriented 3D.
However, I'm not really convinced about this point since the object could have different
I hope author provides additional experiments to rebut this comment.

**Documentation:**

Clear.

**Ethics:**

No ethical problems expected.

**Limitations:**

Their manual efforts might cause errors while annotating objects. However, this is not really a crucial concern since manual annotation is necessary for accurate annotation.

**Opportunities For Improvement:**

There are several points to be improved.

- It is quite hard to follow the entire pipeline since the work involves many processes for dataset creation. The paper would be more readable if it included a figure describing the overall pipeline.
- For those who are not familiar with pose estimation of objects, Section 4.1 does not provide sufficient details of the experiments. What is the input to foundation models? How is the viewpoint encoded with three parameters?
- DINO and MAE are not only trained with object-level images but also with a tremendous number of real-world images. Thus, they might show a weak ability to understand 3D for single object images since they have not seen many of them during training. It would be better if the authors first fine-tuned DeIT, MAE, and others on a 2D-3D dataset such as Objaverse, then checked the 3D awareness.
- What is the major advantage of this ImageNet3D compared to Objaverse? Objaverse also provides rich 3D annotations with sufficiently high quality. The authors noted that they provide canonically aligned 3D, but this canonical orientation is quite ambiguous since objects have different geometries, although they belong to the same category. They should demonstrate that their canonically aligned 3Ds provide useful supervision while training 2D or 3D models.
- Estimation of the quality of natural captions should be added. Readers cannot check how well their proposed caption generation outputs reliable captions.

**Relation To Prior Work:**

Yes.

**Summary And Contributions:**

Existing datasets for 3D object annotations are limited in categories and annotation quality, leading to specialized models that lack generalization. To address this, the authors introduce ImageNet3D, an augmented dataset of 200 categories from ImageNet, providing 2D bounding boxes, 3D poses, 3D locations, and image captions with 3D information. This dataset allows for the analysis of object-level 3D awareness in visual models and the development of general-purpose models for 2D and 3D inference. The work also explores integrating 3D models with language models for 3D-related reasoning, introducing tasks like probing 3D awareness and open vocabulary pose estimation. Experimental results show that ImageNet3D enhances the development of models with robust general-purpose 3D understanding.

---

> ### Author Rebuttal · Authors · 2024-08-17
>
> **`[Weakness 1 & 2]` Clarity of the paper, specifically the dataset creation pipeline, experimental details, and 3D viewpoint parameterization.**
>
> We thank the reviewer for the suggestion. We will aim to improve the presentation of ImageNet3D in our revision, with more visualizations and technical details. Regarding the reviewer’s concern, we provide additional visualizations and explanations as follows:
> 1. We make a figure that provides **an overview of our dataset creation pipeline**. Please refer to `Figure R.1` in the PDF attached in General Response.
> 2. **Experimental details.** To avoid the ambiguity when complex scenes with multiple objects are presented, we adopt a data preprocessing step following previous works [31,33]. Specifically, we crop and resize the images, so the objects are in the center with roughly the same size and very limited background is visible. Hence, during linear probing, models would output image embeddings encoding rich 2D and 3D information of the main object, allowing us to study the object-level 3D awareness of these pretrained models. We will involve more details about our experiments in our revision.
> 3. **Viewpoint parameterization.** We follow PASCAL3D+ and ObjectNet3D and parameterize the 3D viewpoint in a horizontal coordinate system with three parameters: azimuth, elevation (or altitude), and in-plane rotation. We make an illustration of this parameterization as visualized in `Figure R.2` in the PDF attached in General Response. We will add these technical details to our revision.
>
> We are dedicated to improving the presentation of our paper to accommodate future readers from various backgrounds. Please let us know if there are other areas for improvement.
>
> &nbsp;
>
> **`[Weakness 3]` DINO and MAE are not only trained with object-level images but also with a tremendous number of real-world images.**
>
> We agree with the reviewer that visual foundation models such as DINO, MAE, and CLIP are not specialized for object-level 3D awareness. Their limited 3D awareness can be attributed to the training data (as pointed out by the reviewer), as well as their training objectives. However, we note that the value of these visual foundation models is their transferability to a wide range of applications. Similar to previous linear probing experiments on ImageNet-1k classification and low-level 3D awareness [22], we aim to study specific properties of these models while retaining other intriguing properties, such as rich semantics and part correspondence. Our linear probing experiments investigate the object-level 3D awareness of visual foundation models and provide insights when deploying these models for other tasks. For instance, the weak object-level 3D awareness of CLIP explains why vision-language models using CLIP as the visual encoder struggle with questions that require 3D knowledge. While finetuning these models on object-centric data would probably demonstrate better object-level 3D awareness, the finetuned models may not suit broader interests, such as DINO for scene understanding and CLIP as the visual encoder for vision-language models.
>
> &nbsp;
>
> **`[Weakness 4 - Part 1]` Major advantage of ImageNet3D compared to Objaverse.**
>
> We conjecture this is a typo and the reviewer is meant to ask about the advantage over ObjectNet3D. Here we will provide comparisons to both Objaverse and ObjectNet3D.
> 1. **Objaverse.** While Objaverse provide a large number of object instances with diverse shapes and appearances, training 3D vision models on Objaverse are hindered by two challenges: (i) 3D vision models trained on synthetic data would not generalize well to real data, and (ii) 3D models in Objaverse are not canonically aligned, which leads an open research problem of how to utilizing these noisy data [A].
> 2. **ObjectNet3D.** Our advantages over ObjectNet3D are as follows: (i) the 3D annotations in ObjectNet3D have limited 3D quality and we produce more accurate 3D annotations from an iterative process of reviewing and refining (see Sec. E), (ii) our ImageNet3D includes more categories, object instances, and more types of annotations, (iii) with the new annotations in ImageNet3D, we study new problems of how to develop unified 3D vision models (L29-45) and new tasks, which will benefit downstream applications in robotics and vision-language. Note that these new tasks are only made possible in ImageNet3D as we involve most of the common rigid categories in real life, providing a comprehensive study of object-level 3D awareness and how 3D recognition models generalize to unseen categories with semantically similar parts (see Fig. 5).
>
> [A] Liu et al. DIRECT-3D: Learning Direct Text-to-3D Generation on Massive Noisy 3D Data. In CVPR, 2024.
>
> &nbsp;
>
> **`[Weakness 4 - Part 2]` The authors should demonstrate that their canonically aligned 3Ds provide useful supervision while training 2D or 3D models.**
>
> We thank the reviewer for the suggestion. Please refer to `[General Response 1]` for our ablation study results on cross-category 3D alignment. In summary, cross-category 3D alignment (i) is a crucial design to study open-vocabulary 3D pose estimation or unified 3D vision models designed for all rigid categories, and (ii) would lead to improved benchmark performance as models can learn semantics that are shared between categories and benefit from joint training.

---

> > ### Author Rebuttal · Authors · 2024-08-17
> >
> > **`[Weakness 5]` Estimation of the quality of natural captions should be added.**
> >
> > We thank the reviewer for the constructive suggestion but we are not aware of methods to fairly assess the quality of these captions interleaved with 3D annotations. We would like to make further clarifications on this:
> > 1. **The quality of the captions in describing the image.** As we closely follow the caption generation pipeline in LLaVA, we believe that our captions have a reasonable quality summarizing key objects and concepts in the image. This is supported by the results demonstrated in the LLaVA paper -- when prompting GPT-4 with 2D bounding boxes, the captions can be valuable assets to train vision-language models.
> > 2. **The value of the captions when interleaved with 3D tokens.** Captions interleaved with 2D/3D tokens enable a new interface for vision-language models to interact with human pose [25], segmentation maps [26], and object 6D pose as in our work. This is a fairly new problem and evaluation of natural captions with 2D/3D tokens or vision-language models with 2D/3D tokens is an active research problem. Prior work [25] only evaluated on standard pose estimation and failed to show that training on their data would lead to models with improved 3D understanding (rather than simply a better pose estimation model). Many concerns are left unaddressed, e.g., can these 3D-VLMs answer 3D-related questions better with explicit 3D representations? Hence fairly estimating the quality of the captions is a challenging research problem. For future work we hope to build stronger vision-language models with better 3D awareness and present promising evaluations to support these findings.

---

> > > ### Comment · Reviewer_rBvX · 2024-08-29
> > > **Concerns Addressed.**
> > >
> > > Thanks. After rebuttal,
> > > 1. The paper presentation is much more improved so that the paper become much more readable.
> > > 2. Many advantages of ImageNet3D over Objaverse are clearly stated in the rebuttal.
> > > 3. Canonically aligned 3Ds are having many benefits over previous 3D dataset as described in the General Response 1.
> > >
> > > So I'd like to raise my score to the acceptance.

---

> > ### Author Response · Authors · 2024-08-29
> > **Official Comment by Authors**
> >
> > Dear Reviewer rBvX,
> >
> > We are glad to learn that our rebuttal addressed your concerns. Thanks again for your time and constructive suggestions. We will integrate these modifications into our revision and further polish the presentation. Please let us know if there are any further questions.

---

### Official Review · Reviewer_D8US · 2024-07-24
**Resolve the old issues of the traditional 3D object datasets**

**Rating:** 8
**Confidence:** 5
**Correctness:** yes
**Clarity:** yes

**Review:**

The paper is well organized and easy to follow. The released dataset is well structured thus easy to use.

**Strengths:**

The paper introduce a large-scale 2D-3D aligned object dataset, with various annotations such as 3D pose, category and 3D shape, etc, which can accelerate the combined utilization of the dataset with LLMs. Above all, the proposed dataset has a contribution in the aspect of general-purpose 3D object understanding. Even I also had experiences of using 2D-3D alignment dataset such as Pascal3D or ObjectNet3D, and I felt the extreme limitation for the general-purpose 3D orientation understanding on these existing dataset, because there orientation alignment do not consider the 'cross-category', which makes it difficult to achieve generalized orientation or shape estimation. The proposed dataset resolve this issue by using canonical poses and cross-category alignment.

**Additional Feedback:**

None

**Documentation:**

yes

**Limitations:**

Already discussed above

**Opportunities For Improvement:**

Similar to traditional datasets, ensuring the quality of 3D annotations for 2D images remains challenging. For instance, in datasets like Pascal3D or ObjectNet3D, it is often observed that the actual 3D models displayed in 3D space may not always align well with the corresponding 2D images. This limitation likely arises because the alignment of 3D objects is manually estimated based on 2D images without true 3D information.

To enhance the value of large-scale datasets, it would be beneficial to refine the alignment process using geometric estimates from depth image estimations. Reporting on the improvement in alignment accuracy achieved through such methods would also be valuable.

Additionally, it would be advantageous to have a mechanism to customize the canonical pose for each object category. Since the definition of a canonical pose can vary based on subjective criteria, allowing manual adjustment of canonical poses with consideration for cross-category consistency would be beneficial.

**Relation To Prior Work:**

yes

**Summary And Contributions:**

This paper presents a dataset for 3D object understanding from 2D image. The proposed dataset mainly handles the issues such as 3D orientation, localization, category, caption and visual quality of the objects in 2D images.

---

> ### Author Rebuttal · Authors · 2024-08-17
>
> **`[Weakness 1]` Ensuring the quality of 3D annotations for 2D images is challenging. To enhance the value of large-scale datasets, it would be beneficial to refine the alignment process using geometric estimates from depth image estimations.**
>
> We thank the reviewer for this constructive suggestion. This is a very interesting idea to explore, which will both improve the quality of 3D annotations and advance object pose estimation models by exploiting visual foundation models, such as DepthAnythingV2. However, our previous results show that this approach is only successful if the shape is roughly reconstructed when estimating the 3D viewpoint, which benefits tasks such as 3D human pose estimation. For rigid objects with diverse shapes considered in our dataset, we currently don’t have positive results. We find that with a fixed template mesh, i.e., without an approximate shape reconstruction, the depth and surface normals wouldn’t necessarily help refining the annotated object pose. We are actively exploring such methods and will integrate 3D annotation refinements into our dataset if we had success in the future.
>
> &nbsp;
>
> **`[Weakness 2]` It would be advantageous to have a mechanism to customize the canonical pose for each object category.**
>
> We thank the reviewer for this constructive suggestion. We believe this design will benefit research in different areas and increase the impact of our dataset. We will integrate this mechanism in our GitHub codebase.

---

### Official Review · Reviewer_kP2A · 2024-07-24
**Review: ImageNet3D**

**Rating:** 6
**Confidence:** 4
**Correctness:** Yes
**Clarity:** Yes

**Review:**

Pros:

   1. This paper has expanded the amount of image data with 3D annotations. Compared with previous works such as ObjectNet3D, it contains more categories, more information, and more samples. 3D annotation on real data is often expensive, so this work is expected to benefit related fields.
   2. The proposed natural captions may help with LLM-related applications. In addition, the cross-category alignment seems to offer a better canonical space. Although it may still bring problems (listed in Cons), this part is of appropriate novelty.
  3. The writing is easy to follow, and the experiment details are clear to me.

Cons:
   1. Intuition and ablation study on cross-category 3D alignment: This paper tries to address the alignment issue, which is good. However, although certain aspects are considered, they are not guaranteed to benefit the experiments. Some ablation studies will be helpful in such cases, like jittering the canonical pose for each category.
   2. Evaluation: The baseline evaluation in this paper seems quite insufficient. Specifically, although the dataset provides annotations including “6D pose, captions, object visual quality,” only the category and 3D rotation are explored in the experiments. This is even fewer than the 2016 baseline ObjectNet3D, which further explored the bounding box regression. The results with existing baselines are also missing, like PoseCNN [1].
   3. Model design: The linear probing model is counterintuitive since the image can be highly composited (like the 4th image on the 1st row of Figure 1), and no bounding boxes or masks are provided for pose estimation. In contrast, such regional information has been crucial in previous works like PoseCNN and NOVUM. Thus, the performances of different pretrained models can be misjudged, as well as the result of the scaling experiments.

[1] Xiang Y, Schmidt T, Narayanan V, et al. Posecnn: A convolutional neural network for 6d object pose estimation in cluttered scenes. arXiv preprint arXiv:1711.00199, 2017.

**Strengths:**

This paper proposes a more extensive 3D annotated dataset. Although the scale is not significantly larger, it provides additional labels and could benefit the 3D vision community.

**Additional Feedback:**

No

**Documentation:**

Yes

**Limitations:**

Yes, the limitations are discussed.

**Opportunities For Improvement:**

Additional Ablation studies on cross-category alignment, correct explanations, or experiments on evaluation and modeling could improve my rating.

**Relation To Prior Work:**

Yes

**Summary And Contributions:**

This paper introduces a new 3D annotated dataset, augmenting 200 categories from ImageNet with general 3D object attributes. They propose a cross-category 3D alignment to benefit multi-category training. In addition, natural captions are provided for language-related downstream tasks. Although the experimental part is not sufficient, this work demonstrates improvements compared with previous benchmarks.

---

> ### Author Rebuttal · Authors · 2024-08-17
>
> **`[Weakness 1]` Cross-category 3D alignment is not guaranteed to benefit the experiments. Some ablation studies will be helpful in such cases.**
>
> We thank the reviewer for the suggestion. Please refer to `[General Response 1]` for our ablation study results on cross-category 3D alignment. In summary, cross-category 3D alignment (i) is a crucial design to study open-vocabulary 3D pose estimation or unified 3D vision models designed for all rigid categories, and (ii) would lead to improved benchmark performance as models can learn semantics that are shared between categories and benefit from joint training.
>
> &nbsp;
>
> **`[Weakness 2]` Baseline evaluation in this paper seems insufficient compared to ObjectNet3D.**
>
> We thank the reviewer for the suggestion. Please refer to `[General Response 2]` for our new baseline results on 3D pose estimation and 2D object detection.
>
> We would like to elaborate our choices of experiment tasks. Recent vision models have demonstrated improved performance on a range of 2D tasks, such as classification, detection, and segmentation, with the advancements of large-scale pretraining. Meanwhile, there are limited studies of how to improve the object-level 3D understanding of vision models. This is largely due to the lack of a large-scale dataset with high-quality 3D annotations to train strong 3D vision models or to evaluate the object-level 3D awareness of existing models. Therefore, we develop the ImageNet3D dataset and hope to fill in this gap. In our submission, we focus on 3D tasks only, as there are many well-developed benchmarks for 2D vision tasks, such as classification and object detection. However, we agree with the reviewer that these 2D baseline results will help to understand the annotation quality of ImageNet3D and to benefit future studies. We will add these new baseline evaluation results to our revision.
>
> &nbsp;
>
> **`[Weakness 3]` The linear probing model is counterintuitive since the image can be highly composited and no bounding boxes or masks are provided for pose estimation.**
>
> We apologize for the confusion, as we didn’t include some important technical details of our experiments in the submission. To avoid the ambiguity when complex scenes with multiple objects are presented, we adopt a data preprocessing step following previous works [31,33]. Specifically, we crop and resize the images, so the objects are in the center with roughly the same size and limited background is visible. Hence, during linear probing, models would output image embeddings encoding rich 2D and 3D information of the main object, allowing us to study the object-level 3D awareness of these pretrained models. We hope this addresses the reviewer’s concern and justifies the legitimacy of our experimental results.
>
> Please refer to `Figure R.3` in our attached PDF for some qualitative examples of our data pre-processing. The data pre-processing code can also be found in our released repo [here](https://github.com/wufeim/imagenet3d_exp/blob/main/scripts/preprocess_data.py#L27).

---

> > ### Comment · Reviewer_kP2A · 2024-08-30
> >
> > The authors’ rebuttal has addressed most of my concerns, and I improve my rating from 5 to 6. Please include the new experiments and experimental details in the revision.

---

> > ### Author Response · Authors · 2024-08-31
> > **Official Comment by Authors**
> >
> > Dear Reviewer kP2A,
> >
> > We are glad to hear that our rebuttal addressed your concerns.
> >
> > &nbsp;
> >
> > **Regarding the PoseCNN baseline:** The original PoseCNN was proposed for instance-level pose estimation, which assumes the detailed appearance and shape to be known. Our early experiments that directly adapt PoseCNN to category-level pose estimation on ImageNet3D fail as instances in ImageNet3D from a certain category would vary in both shape and appearance. We extend the PoseCNN with category template meshes and force the model to learn pose representations given various shapes and appearances. We don't have the full results yet but the results on 12 categories in ImageNet3D (same categories as in PASCAL3D+) are reported below. We will present the full results of PoseCNN in our revision.
> >
> > | Model | Joint classification and pose estimation |
> > |:-|:-:|
> > | ResNet-50 | 73.33 |
> > | SwinTransformer-Tiny | 74.67 |
> > | NMM-Sphere | 76.69 |
> > | PoseCNN | 70.10 |
> >
> > &nbsp;
> >
> > Thanks again for your time and constructive suggestions. We will add the new baselines results on object detection and pose estimation to our revision. Please let us know if there are any further questions.

---

### Author Rebuttal · Authors · 2024-08-17

We would like to thank all the reviewers for their valuable comments and suggestions. We are encouraged by the positive feedback, finding our dataset “beneficial to related fields”, our writing “easy to follow”, and our released code and dataset “easy to use”. In the following we respond to common questions and concerns from the reviewers, and will include these in our future revisions.

&nbsp;

**`[General Response 1]` Benefits of cross-category 3D alignment. (kP2A, rBvX)**

A common concern by reviewers is the benefit of cross-category 3D alignment adopted in ImageNet3D. Specifically we align the canonical poses of all 200 categories in ImageNet3D based on shared semantic parts between different categories or similar shapes (see Fig. 2). Meanwhile this design is missing in previous datasets such as ObjectNet3D (see Fig. 3). As suggested by Reviewer D8US, we will further integrate an option in our codebase for users to customize the canonical poses that suit their needs.

In general, there are two benefits of cross-category 3D alignment:
1. **Cross-category 3D alignment provides a pre-defined canonical pose shared by all rigid categories, seen or unseen during training.** This is crucial for the task of open-vocabulary pose estimation presented in our work -- without cross-category 3D alignment, models wouldn’t generalize to novel categories with a different canonical pose (consider the cases in Fig. 3). Moreover, developing an unified 3D model for all rigid categories would be an ill-posed problem because models can fit to the canonical poses of seen categories during training, but would fail to generalize due to the ambiguity of canonical poses of novel categories.
2. **Cross-category 3D alignment allows models to learn shared semantics between different categories, yielding a higher benchmark performance.** We follow the suggestion of reviewer kP2A and randomly jitter the canonical poses of 1/3 of the categories and compare the performance before and after cross-category 3D alignment. The quantitative results are reported below and we can see that all models achieve better results when cross-category 3D alignment is adopted. This shows that models can learn semantics that are shared between categories and achieve higher performance from joint training. This further demonstrates the importance of cross-category 3D alignment when developing unified 3D vision models.

| Model | w/ cross-category 3D alignment | avg | electronics | furniture | household | music | sports | transportation | work |
|:-|:-|:-:|:-:|:-:|:-:|:-:|:-:|:-:|:-:|
| ResNet50 | no | 47.6 | 56.9 | 63.0 | 40.0 | 39.0 | 27.1 | 59.3 | 32.0 |
| ResNet50 | yes | 50.9 | 60.0 | 67.2 | 43.0 | 43.8 | 27.7 | 64.1 | 33.8 |
| SwinTrans-T | no | 49.8 | 60.0 | 67.0 | 42.2 | 43.6 | 29.6 | 60.5 | 32.6 |
| SwinTrans-T | yes | 53.2 | 63.1 | 71.6 | 44.8 | 45.3 | 30.4 | 66.2 | 35.0 |
| NMM-Sphere * | no | 10.6 | 3.8 | 12.1 | 6.7 | 1.2 | 2.1 | 26.0 | 4.2 |
| NMM-Sphere | yes | 57.4 | 61.3 | 65.9 | 52.4 | 51.7 | 40.5 | 67.9 | 43.4 |

\* Here NMM-Sphere fail to converge well without cross-category 3D alignment. The reason is that misaligned canonical poses lead to false negative pairs and break the part contrastive learning. For instance, if "shoe" and "slipper" have misaligned canonical pose, then the shared semantic parts would form negative pairs in contrastive learning. Instead, shared semantic parts would form positive pairs and produce similar part embeddings when cross-category 3D alignment is adopted.

&nbsp;

**`[General Response 2]` Additional baseline results on 3D pose estimation and 2D object detection. (kP2A)**

We thank the reviewer for the suggestions and provide additional baseline evaluations:
1. **2D object detection.** We consider a CNN model (i.e., Faster R-CNN) and a transformer model (i.e., DETR) as the baselines for 2D object detection. We follow the implementation and hyperparameters from mmdetection. Due to the limited time to prepare this rebuttal, we weren't able to evaluate more baseline models on this task. We will involve more baseline evaluations in our revision (e.g., Sparse R-CNN and ViTDet).
2. **3D pose estimation.** As suggested by Reviewer kP2A, we will add PoseCNN as a baseline of our pose estimation tasks. However, the PoseCNN codebase is a bit outdated and requires certain modifications to accommodate different data formats and to adapt the model for category-level pose estimation. Given limited time, we weren’t able to reproduce a reasonable performance for PoseCNN. **We will update the results of PoseCNN during the discussion period.**

| Model | box AP |
|:-|:-:|
| Faster R-CNN | 38.9 |
| DETR | 42.5 |

&nbsp;

**`[General Response 3]` Additional results on models’ performance given different training time (pu6E).**

We analyze the computational costs of training joint classification and pose estimation models on our ImageNet3D with 200 categories and on a subset of ImageNet3D with 100 categories (same categories as ObjectNet3D). We visualize the performance by clock time in `Figure R.4` of the attached PDF.

Results show that for CNN or transformer models that formulate pose estimation as a classification problem (i.e., ResNet or SwinTransformer), they generally converge pretty fast and take a similar amount of time to converge on ImageNet3D or the 100-class subset. These methods don't require many computational resources to train.

Meanwhile neural mesh models (i.e., NMM-sphere) take much longer to train and converge. This is because these models learn part-contrastive features with contrastive losses, which limit the training batch size and generally don’t converge as fast as standard classification objectives. For future work we will explore methods to improve the training efficiency of neural mesh models, such as by involving better foreground/background feature sampling methods or by mining hard-negative pairs.

---

### Author Response · Authors · 2024-08-25
**Looking forward to further discussions**

Dear Reviewers,

Thank you again for your valuable time and constructive suggestions. For each of the concerns, we provided extra technical details, experimental results, or detailed discussions. We hope these results help to clarify certain aspects of our dataset and may address your concerns.

We will incorporate several suggested changes into our revision and hopefully will improve the quality and impact of our dataset. Meanwhile there are a few open questions/discussions raised by the reviewers. We hope to hear more about your opinions and see if our answers addressed your concern.

Thanks again for your time. If there are any other questions about our dataset, we are more than willing to provide further clarifications.

---

### Decision · Program_Chairs · 2024-09-26

**Decision:**

Accept (Poster)

**Comment:**

The submission initially received mixed reviews; post-rebuttal, all reviewers became positive and recommended acceptance. The AC concurs.  Please revise the submission accordingly for the camera ready.